# A Sizer model for cell differentiation in *Arabidopsis thaliana* root growth

Irina Pavelescu[1,2], Josep Vilarrasa-Blasi[1,‡], Ainoa Planas-Riverola[1], Mary-Paz González-García[1,§], Ana I Caño-Delgado[1,†,*] (iD) & Marta Ibañes[2,3,†,**] (iD)

## Abstract

Plant roots grow due to cell division in the meristem and subsequent cell elongation and differentiation, a tightly coordinated process that ensures growth and adaptation to the changing environment. How the newly formed cells decide to stop elongating becoming fully differentiated is not yet understood. To address this question, we established a novel approach that combines the quantitative phenotypic variability of wild-type *Arabidopsis* roots with computational data from mathematical models. Our analyses reveal that primary root growth is consistent with a Sizer mechanism, in which cells sense their length and stop elongating when reaching a threshold value. The local expression of brassinosteroid receptors only in the meristem is sufficient to set this value. Analysis of roots insensitive to BR signaling and of roots with gibberellin biosynthesis inhibited suggests distinct roles of these hormones on cell expansion termination. Overall, our study underscores the value of using computational modeling together with quantitative data to understand root growth.

**Keywords** *Arabidopsis* root zonation; brassinosteroids; cell differentiation; computational analysis; phenotypic variability

**Subject Categories** Development & Differentiation; Plant Biology; Quantitative Biology & Dynamical Systems

**Mol Syst Biol. (2018) 14: e7687**

## Introduction

Root growth is essential for plant survival and adaptation to environmental stresses. Most of our current understanding of root growth and development was derived from studies on the model species, *Arabidopsis thaliana* (*Arabidopsis*). Its primary root displays a characteristic architecture consisting of cell files arranged as concentric circles, with the stem cell niche and its quiescent center (QC) located at the inner site of the root apex (Dolan *et al*, 1993; Sarkar *et al*, 2007). The primary root is divided into three distinct and consecutive developmental zones along its longitudinal axis (Dolan *et al*, 1993). The three developmental zones reflect the temporal evolution of cells within roots, where cells grow and divide by the same principles. In the zone closest to the QC, known as the meristem (MZ), cells divide upon expansion. In the next developmental zone, the elongation zone (EZ), cells elongate and gradually differentiate without further cell division. The third zone, the differentiation zone (DZ), consists of mature, terminally differentiated, cells that no longer elongate. Morphologically, the DZ in the wild type (WT) is marked by the appearance of epidermal root hairs on the external surface of outer cell files (trichoblast) and the existence of fully differentiated xylem for internal cell files (Dolan *et al*, 1993; Ishikawa & Evans, 1995; Beemster *et al*, 2003; Verbelen *et al*, 2006; Zhang *et al*, 2010; Mähönen *et al*, 2014).

The root growth is driven by the division of meristematic cells in the root apex and their subsequent cell elongation in the proximal side of the meristem. Root growth can reach a stationary regime that is the result of a fine balance between proliferation, elongation, and differentiation (Beemster & Baskin, 1998; Verbelen *et al*, 2006; Ivanov & Dubrovsky, 2013; Takatsuka & Umeda, 2014). In the stationary phase of root growth, the sizes of the MZ and EZ remain constant, and the primary root grows proportionally with time by increasing the length of its DZ. In this regime, the root growth is mostly dictated by how long mature cells are and how often the meristem proliferates. To understand stationary root growth, it becomes necessary to know how mature cell length and meristematic activity are set and coupled, and hence, how root zonation proceeds.

The cellular and molecular analysis of different root zones in WT and mutant plants has dramatically increased our understanding of root growth and development. Genetic screening for mutants with deficient root growth identified several regulators related to root cell

1   Department of Molecular Genetics, Center for Research in Agricultural Genomics (CRAG), CSIC-IRTA-UAB-UB, Campus UAB, Bellaterra (Cerdanyola del Vallès), Barcelona, Spain
2   Departament de Física de la Matèria Condensada, Universitat de Barcelona, Barcelona, Spain
3   Universitat de Barcelona Institute of Complex Systems (UBICS), Universitat de Barcelona, Barcelona, Spain
    *Corresponding author. Tel: +34 93 563 66 00 Ext. 3210; Fax: +34 93 563 66 01; E-mail: ana.cano@cragenomica.es
    **Corresponding author. Tel: +34 93 403 91 77; E-mail: mibanes@ub.edu
    †These authors contributed equally to this work
    ‡Present address: Carnegie Institution for Science, Department of Plant Biology, Stanford, CA, USA
    §Present address: Centro Nacional de Biotecnología-CSIC, Madrid, Spain

division and elongation (Benfey *et al*, 1993; Li *et al*, 2001; Mouchel *et al*, 2004; Rodrigues *et al*, 2009). Mutant plants with defects in root growth typically have root zonation altered (Mouchel *et al*, 2004; Ubeda-Tomás *et al*, 2009; González-García *et al*, 2011; Hacham *et al*, 2011; Meijón *et al*, 2014). Compiling data support that the transition from the meristem to the EZ is positional information-driven (named Ruler mechanism, hereafter, according to the nomenclature in De Vos *et al*, 2014 in the context of root zonation). In the Ruler mechanism, the developmental decision for zonation is based on the spatial position of the cells, with this positional information typically conferred by signaling gradients (Grieneisen *et al*, 2012). This is widely supported by the role of auxin, a phytohormone, and the auxin-induced transcription factors *PLETHORA* in the establishment of MZ size (Aida *et al*, 2004; Galinha *et al*, 2007; Grieneisen *et al*, 2007; Mähönen *et al*, 2014). Also, computational modeling has shown that a Ruler mechanism may account for coherent root growth between root files (De Vos *et al*, 2014). Other hormones, such as brassinosteroids (BRs), cytokinins, and gibberellins (GA), crosstalk with auxin signaling and thereby contribute to the size of root meristems (Liu *et al*, 2014; Sozzani & Iyer-Pascuzzi, 2014).

Collectively, these studies advance our understanding of how cells transition from division with expansion to elongation during primary root growth. However, the regulatory mechanisms that control the transition from elongation (EZ) to differentiation (DZ) have received far less attention and remain poorly understood. We describe three different hypotheses that can be raised based on data. First, the robustness of the average time cells spend elongating to many different hormone treatments has lead to the hypothesis that cells elongate during a fixed amount of time and thus stop elongation when reaching this robust time (Beemster & Baskin, 1998), what we name hereafter as the Timer mechanism. We use this terminology because of its analogy with the Timer mechanism in the cell-cycle field, according to which a growing cell performs the next cell-cycle event, like cell division, after a constant time has passed (Campos *et al*, 2014; De Vos *et al*, 2014; Amodeo & Skotheim, 2016). In the context of the root, elongating non-dividing root cells could potentially measure time through the accumulation of a molecular component, for instance, as computationally modeled in (Mähönen *et al*, 2014). Second, a mechanism in which cells stop elongating when reaching a defined length could be hypothesized. Hereafter, we term such a mechanism a Sizer, because of its correspondence to the Sizer mechanism described in the cell-cycle context (Campos *et al*, 2014; Amodeo & Skotheim, 2016). In the cell-cycle field, it has been investigated whether cells divide when reaching a threshold size, which is termed the Sizer. Although a Sizer type of mechanism for terminating cell elongation in roots has been used when modeling the root (Grieneisen *et al*, 2007), its plausibility has not been tested directly. It has been proposed that dilution of gibberellin signaling by cell expansion can terminate cell elongation (Band *et al*, 2012). Such a molecular mechanism could fit with a Sizer mechanism, since the level of gibberellin signaling is thus dependent on the length of the cell. However, the recent visualization of gibberellins suggests another type of gradient (Rizza *et al*, 2017). Third, across natural variation, the length of the mature cell in the root exhibits a correlation with the length of the meristem (Meijón *et al*, 2014), suggesting a possible dependence of these elements and a Ruler type of mechanism to set the EZ.

The pleiotropic defects exhibited by known mutants and hormone-treated roots do not help to decipher which of these three (Timer, Sizer, or Ruler) mechanisms to stop cell elongation is actually occurring in roots. For instance, the short root phenotype of mutants insensitive to BRs (*bri1-116* and *bri1-2*), with mutations in the BR receptor transmembrane kinase brassinosteroid insensitive1 (BRI1), entails reduced meristematic activity and short mature cell lengths, as well as reduced lengths of MZ and EZ (González-García *et al*, 2011; Hacham *et al*, 2011; Cole *et al*, 2014). Additional characterization of *bri1-2* showed decreased average cell elongation rate, but unaltered average time cells spend elongating compared to the WT (Cole *et al*, 2014). This pleiotropic phenotype can be explained by either a Ruler, a Timer, or a Sizer mechanism (see Appendix Text: Section S0), yet the role of BR signaling depends on which mechanism for terminating cell elongation underlies root growth. For instance, by assuming root growth operates under a Timer mechanism, BR signaling would not control what triggers cells to stop elongating (Time), although it would regulate root growth by modifying cell growth only, by impinging just in the meristematic cell division and cell elongation processes, as recently suggested (Kang *et al*, 2017). In other words, in these BR insensitive mutants cellular defects such as a reduced mature cell length would be a mere consequence of the defects in cell growth. Thus, according to the Timer model, BRs would not directly participate on the mechanism that dictates termination of cell elongation. Another possible scenario is that root growth proceeds through a Sizer mechanism. In this case, BRs would participate in the mechanism that dictates termination of expansion besides controlling cell growth (i.e., cell elongation rate and meristematic activity). Therefore, it remains necessary to unveil which mechanism is terminating elongation in the WT, as well as in the mutant, to propose the role of BR signaling.

The present study combines computational and experimental approaches to investigate how root cells terminate elongation and to pinpoint the role of BR signaling in this process. Using mathematical and computational methods, together with quantitative empirical data, we focused specifically on testing the Ruler, Timer, and Sizer mechanisms. The computational results show that the three mechanisms are plausible and drive similar root growth that is consistent with wild-type *Arabidopsis* root growth. However, our mathematical and computational analyses indicate that each mechanism can be distinguished at the quantitative level by relationships between specific pairs of phenotypic traits. The intrinsic quantitative variability of phenotypic traits among isogenic *Arabidopsis* (Col-0) wild-type roots enables to explore these relationships. Together, the quantitative data support that root epidermal and cortical final cell differentiation is modulated by a Sizer mechanism. Accordingly, we propose that root cells sense their length to terminate elongation. To evaluate further this mechanism, we analyzed roots with reduced mature cell lengths, such as the BR insensitive mutant *bri1-116*. Our analysis supports that the Sizer model, with reduced threshold length, cell elongation rate, and meristematic proliferation, is not sufficient to account for the quantitative data in *bri1-116*. Instead, this mutant is well described by a mix of the Sizer and Timer models. This suggests that BR signaling through BRI1 suppresses the Timer mechanism, which appears to participate in the absence of BRI1-mediated signaling and not in the WT. Thereby, BRI1 signaling facilitates that the termination of elongation proceeds only through the Sizer mechanism in the WT, while increasing the

threshold length, cell elongation rate, and meristematic activity. Moreover, we found that BRI1 receptor only at the dividing cells is sufficient to control the mature cell length and overall root growth. Finally, we show that the growth of plants chemically inhibited for the biosynthesis of gibberellin, which are known to have short mature cell lengths (Ubeda-Tomás *et al*, 2008, 2009; Band *et al*, 2012), is consistent with the Sizer mechanism, like in the WT, but with decreased threshold length, elongation rate, and meristematic activity. Therefore, the results suggest that both gibberellin and BR signaling, which are known to crosstalk (Ross & Quittenden, 2016), participate in setting cell expansion termination, although in very distinct manners. Together, our results provide a comprehensive systems approach for dissecting stationary root development and suggest a role for the Sizer mechanism during elongation and cell differentiation in roots.

## Results

### A model for cell elongation dynamics during stationary root growth

To investigate which mechanism drives cells to stop elongating and become fully differentiated mature cells, we first constructed a computational model for stationary root growth dynamics in the EZ and DZ. Based on the architecture of the root, the model considered cells within a single file in these two zones (Fig 1A and B). It assumed that cells in the EZ can only elongate, without dividing, until becoming fully differentiated. Full differentiation corresponds to cessation of growth and thus incorporation into the DZ (Fig 1B). Because the same type of zonation has been described for both the epidermis and the cortex tissues, the model can be applied to either tissue, using in each case the appropriate tissue parameters.

The effect of the mitotic activity at the meristem was modeled as the addition of new cells, each of length $l_0$, into the EZ (Fig 1B). To take into account the fact that meristematic mitotic events in *Arabidopsis* roots are not periodic and exhibit a certain degree of stochasticity (Mähönen *et al*, 2014), the model considered that consecutive cells enter the EZ at random time intervals, which for simplicity are assumed to follow a Gaussian distribution around an average period (Fig 1B, Materials and Methods). The model also assumed that cells can have slightly distinct cell lengths $l_0$ when entering the EZ, exhibiting a Gaussian distribution around an average cell length (Materials and Methods).

In the EZ of *Arabidopsis* roots, cells elongate up to more than ten times their length at the MZ in 6–8 h through a complex mechanical process that involves interactions between cell files. Despite its complexity, exponential elongation over time with a relative rate of cell elongation that is mostly constant fits appropriately quantitative data on increasing cell lengths along the EZ (Band *et al*, 2012; Cole *et al*, 2014). Therefore, we modeled cell elongation as an exponential growth with a constant relative cell elongation rate over time (Fig 1B, lower panel). Cells of the same root file were set to elongate with slightly different, yet constant, relative cell elongation rates chosen at random according to a Gaussian distribution (Materials and Methods). As it will be shown, the choice of exponential growth dynamics is not essential for two of the mechanisms of cell elongation termination.

In addition, cells from distinct roots were assumed to exhibit a larger random variability than cells within the same root file (Materials and Methods). Notice that our approach sets the meristematic activity, the cell elongation rate, and the length of cells when entering in the EZ as independent parameters. Thus, the variability these traits exhibit for the same genotype (but not necessarily between distinct genotypes) is assumed to be independent of each other.

To evaluate whether the model is sufficient to describe qualitatively root growth dynamics in the EZ, we simulated different roots, of which we only simulated the dynamics along a single cell file. Parameter values were chosen within biological reasonable ranges, which could describe qualitatively either epidermal or cortex cell files of Columbia-0 (Col-0) ecotype wild-type roots (Materials and Methods). Our results showed that, for each simulated root file, the dynamics drive an exponential increase in cell length with respect to cell position along the file (Fig 1C). This is consistent with the behavior previously reported of cortex cells at the EZ (supplementary figure in Cole *et al*, 2014) and is readily expected in the absence of variability between cells and for constant relative cell elongation rates (Cole *et al*, 2014; Appendix Text: Section S1.A). The exponential increase is characterized by what we named the "elongation factor" ($r_{MZ}$, $r_{EZ}$) (Fig 1C), which sets a measure for the average number of times a cell is longer than the adjacent cell located one position closer to the QC at the root apex (Fig 1A).

To confirm, evaluate, and quantify the extent of such exponential behavior in individual plant roots, we analyzed root epidermal (trichoblast, to be able to recognize root hair) and cortex cell files of wild-type Col-0 ecotype, from day 1 to 10 postgermination (Materials and Methods, Dataset EV1). Cells closer to the QC show a rather similar yet slightly increasing size, while cells further away increase strongly in length exponentially (Fig 1C). This exponential profile applies for both epidermis and cortex cell files in individually analyzed roots, even before these reach the stationary growth phase (Fig 1C and Appendix Figs S1 and S2, total $n = 340$, see Table EV1 for $n$ values corresponding to each day and tissue). Therefore, we concluded that the model is sufficient to describe the qualitative spatial profile of cell lengths along the EZ.

Given this general exponential behavior found in WT roots, a new method was set to extract the elongation factor and the number of cells in the MZ and the EZ in each plant root (Appendix Fig S3 and Materials and Methods). This method involved the automatic fitting of data of single root files, each from an individual root, to exponential functions. Criteria were set to select which functions fitted best (see method description in Appendix Text: Section S1.B, graphical visualization and validation in Appendix Fig S3 and its description in Appendix Text: Section S1.C, program code in Appendix Text: Section S3.A). We found no significant difference in the average elongation factor $r_{EZ}$ between epidermis and cortex at day 6 at the EZ, which was $1.29 \pm 0.10$ and $1.31 \pm 0.09$, respectively (Appendix Figs S1 and S2, Table EV2, Dataset EV2). The analysis showed that the MZ and EZ reach the steady state at day 6, as expected (Dello Ioio *et al*, 2008; Moubayidin *et al*, 2010; Appendix Fig S1). The MZ stationary values of number of cells extracted using our method were $26.2 \pm 0.5$ cells (epidermis) and $31.1 \pm 1.3$ cells (cortex). Similarly, the number of cells in the EZ at steady state was $11.8 \pm 0.4$ cells (epidermis) and $10.7 \pm 0.7$ cells (cortex) (Table EV2). See Appendix Text (Section S2.A) for

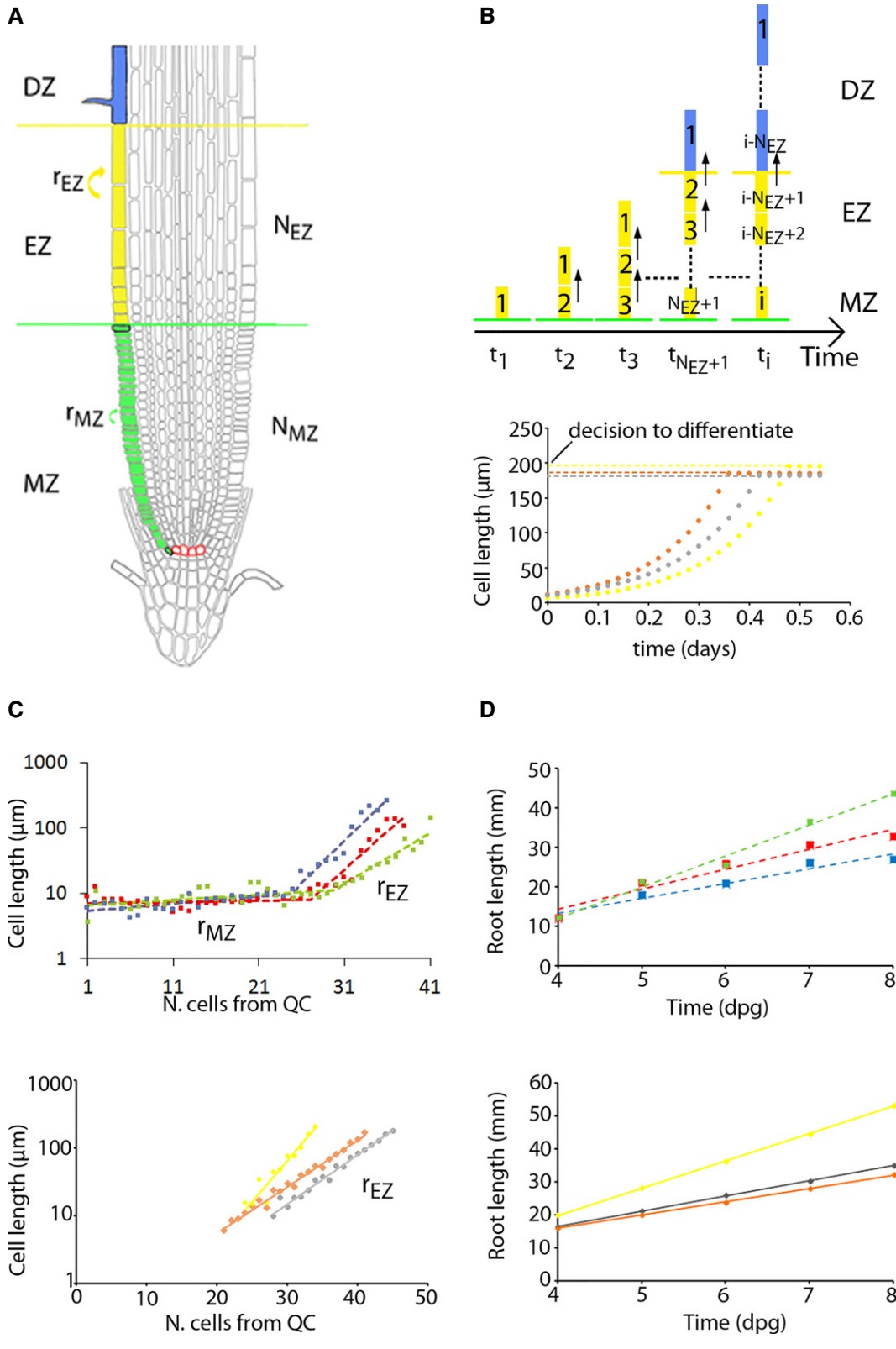

Figure 1.

discussion with previous reported values (Dello Ioio *et al*, 2008; Moubayidin *et al*, 2010; González-García *et al*, 2011; Hacham *et al*, 2011; Mähönen *et al*, 2014).

Overall, these results show that our computational model drives a spatial behavior along the EZ consistent with the one showed by epidermal and cortex cell files in *Arabidopsis* roots. Importantly,

**Figure 1. A model for stationary root growth.**

A   Root cartoon with a cell file colored to depict the MZ (green) with $N_{MZ}$ cells, the EZ (yellow) with $N_{EZ}$ cells, and the DZ (blue), which extends more cells upwards (not shown). $r_{MZ}$ and $r_{EZ}$ are growth factors of each zone. The first and last meristematic cells, and first mature cells in the DZ are encircled in black, while QC cells are in red. The lateral root cap is not entirely depicted for simplicity.

B   Upper panel: Root growth model cartoon. Numbers (i = 1, 2, 3 . . .) label the order at which each cell (rectangle) entered the EZ (yellow) and the DZ (blue). $t_i$ is the time at which the i-th cell entered the EZ. Because of the irregularity in mitotic events at the meristem, $t_i - t_{i-1}$ is set to be random. The simulations start with a single cell in the EZ. The EZ grows until it reaches the stationary regime, when it will be composed of nearly the same number of cells over time. All results in figures correspond to the stationary regime. Lower panel: Simulation of cell length exponential growth for three different cells (colors) of three simulated root files. Results herein correspond to the Sizer model.

C   Cell length profile as a function of the cell number position from the QC for three wild-type seedlings (symbols, top) at day 8 postgermination and three simulated root files for the Sizer model (symbols, bottom). Notice the use of a logarithmic scale in cell lengths, which visualizes the cell length profiles as linear (being exponential in a linear scale). Lines are fittings, and the slope in the EZ corresponds to the logarithm of the elongation factor ($\log_{10} r_{EZ}$, Materials and Methods).

D   Root growth over time for three wild-type seedlings (symbols, top) and three simulated roots (symbols, bottom) as in (C). Lines are linear fittings. In all the plots, arbitrary values of the initial root length and the cell position from the QC for the first elongating cell have been set for the simulated data.

this spatial behavior prompted a new method to set the boundary between the MZ and EZ.

**Three putative mechanisms for terminal cell differentiation**

During the stationary phase of root growth, new cells enter the EZ, while others mature and exit the EZ such that the number of cells in the EZ remains constant. To establish the size of the EZ and model stationary root growth, it is necessary to define what makes cells stop elongating, becoming mature, and entering the DZ. Thus, we modeled three main putative mechanisms of developmental decisions (Ruler, Timer, and Sizer), by defining specific differentiation (i.e., termination of elongation) rules in each case (Fig 2A and B, and Materials and Methods): In the Ruler model, cells stop elongating when they reach a threshold distance from the meristem; in the Timer model, cells stop growing when they have been elongating for a given time; and in the Sizer model, cells stop elongating when reaching a specific threshold length (Fig 2A and B).

As expected, the simulations of the three models confirmed that all drive a linear root growth in the stationary regime, consistent with the stationary growth of *Arabidopsis* roots (Fig 1D and Appendix Fig S4A). Therefore, these three models are all able to account for two main characteristic features of root growth: exponential profile of cell lengths in the EZ and linear root growth with time. However, which of these mechanisms is present in real roots awaits to be uncovered.

**The three models can give distinct quantitative predictions**

We first evaluated whether the three models drive quantitative accurate root growth dynamics by setting realistic wild-type parameter values for cell elongation rate, cell size when entering the EZ, and time between consecutive cells entering the EZ. To this end, we first inferred a dynamical description of wild-type *Arabidopsis* roots in the stationary regime by measuring root lengths over time (Materials and Methods, $n = 20$) and by using our quantification of the spatial cell length longitudinal profiles along the EZ for those days in the stationary regime of root growth (data from day 6, 8, and 10, $n = 100$, for epidermis, data from day 6, $n = 30$, for cortex, Dataset EV2). Inferred average values (Table EV2, Materials and Methods) were consistent with those previously reported in the literature (Beemster & Baskin, 1998, 2000; Cole *et al*, 2014; Mähönen *et al*, 2014; Appendix Text: Section S2.B). In addition, by quantifying individual roots, we extracted a measure of variability between roots for

all phenotypic and inferred dynamical traits (Table EV2; Approach 2; $n = 22$, Dataset EV2). To validate our method to infer these dynamical traits, we applied it to data extracted from simulated root files, for all three models (Appendix Text: Section S1.D and Appendix Figs S5–S7).

The analysis of individual simulated root cell files for each model, using the ranges of relative cell elongation rates, meristematic activities, and cell elongation termination thresholds extracted from WT roots (Materials and Methods), revealed that the three models can be set to account for several phenotypic traits of WT root growth (Appendix Fig S4B–F). Specifically, parameter values within the models were chosen such that the root growth rate ($R_{root}$), mature cell length ($l_{max}$, measured as the length of the EZ cell next to the root hair cell), number of cells and length of the EZ ($N_{EZ}$, $L_{EZ}$), and elongation factor ($r_{EZ}$) of each model ($n = 1,000$) were statistically consistent with those extracted from the WT in the stationary regime (Tables EV3 and EV4, $n = 122$ for the epidermal WT data, corresponding to day 6, 8, and 10 postgermination, Dataset EV2).

Yet, the question remains whether the models drive distinct predictions that can be used to evaluate further the plausibility of each mechanism. Theoretical and numerical analysis of the three models indicated that they can drive different predictions each under changes of the parameters controlling root growth. All models predict the same type of root changes when only the threshold value that sets cell elongation termination changes (Appendix Fig S8 and Appendix Text: Section S1.E). In this case, the only difference is quantitative, with the Timer model being the most sensitive to changes in the threshold time (Appendix Fig S8). However, each model predicts different outcomes when the meristematic activity and/or the relative cell elongation rate change and the threshold for cell elongation termination remains constant. These different predictions correspond to distinct relationships between the phenotypic traits in the EZ (see Appendix Text: Section S1.E for the mathematical expression of these relationships, depicted with lines in Appendix Figs S9 and S10). In this case, the Ruler model predicts no correlation between the number of cells in the EZ and its length (gray line in Appendix Fig S9B). In this model, if cells along the EZ are shorter (i.e., the elongation factor $r_{EZ}$ is smaller because the relative elongation rate decreases and/or the meristematic activity decreases), then the EZ is composed of more cells to fill up the constant threshold distance from the meristem (gray line in Appendix Fig S9D) and consequently mature cells become shorter (gray line in Appendix Fig S9C). In contrast, the Sizer model predicts that the length of mature cells does not depend on how long

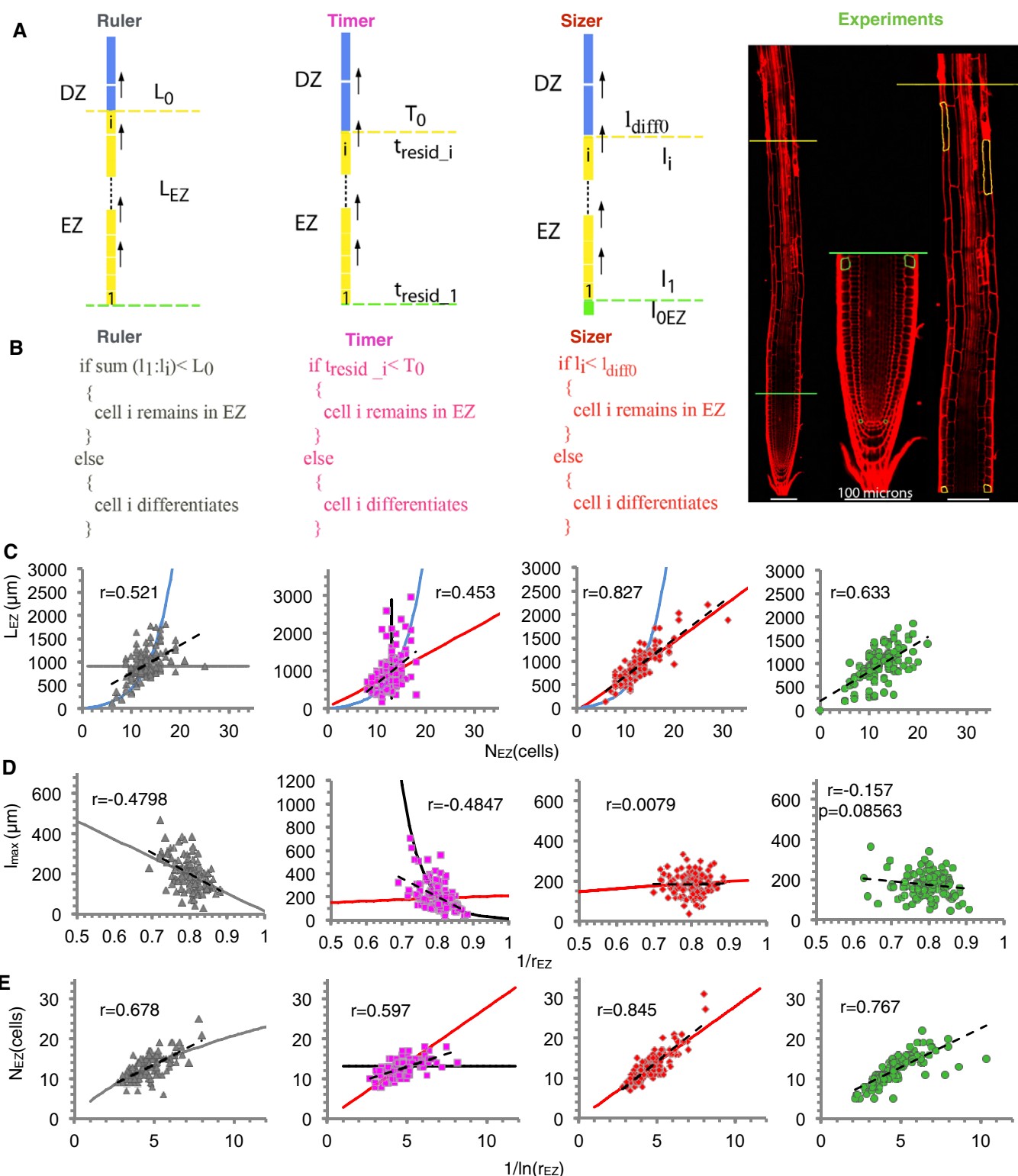

**Figure 2.**

or short are the cells along the EZ (i.e., does not depend on $r_{EZ}$) since the mature cell length is fixed by the constant threshold cell length. Yet, this model predicts that an EZ with shorter cells (i.e., smaller $r_{EZ}$) entails more cells elongating until reaching the threshold mature cell length (red line in Appendix Fig S9D) and thereby

drives larger EZ lengths (red line in Appendix Fig S9B). Because the cell lengths along the EZ exhibit an exponential increase as a function of their cell number position from the QC (Fig 1C, and Appendix Figs S1 and S2), the Sizer model predicts that the number of cells in the EZ (i.e., up to reaching the threshold cell length) is

**Figure 2.    Comparison between the predictions of three models for final differentiation with empirical data from the epidermis in *Arabidopsis thaliana* roots.**

A       Cartoon of each cell terminal differentiation mechanism (Ruler in gray, Timer in pink, and Sizer in red) and (right) juxtaposed (red lines) confocal images of an 8-day-old WT seedling with the zones being indicated. Colors of zones as in Fig 1. Note the differences in subindex nomenclature compared with Fig 1.

B       Pseudocode of the algorithm used in each model.

C–E    Relationships between pairs of phenotypic traits. (C) Length of the elongation zone $L_{EZ}$ versus its number of cells, $N_{EZ}$. (D) Length of mature cell (EZ cell closest to DZ) $l_{max}$ versus $1/r_{EZ}$. (E) $N_{EZ}$ versus $1/\ln(r_{EZ})$, which can be interpreted as the ratio between the meristematic activity ($R_{prod}$) and the elongation rate ($r_{elong}$). Panels from left to right: the Ruler, Timer, and Sizer models, and epidermal Col-0 data. Symbols in the three left-most panels represent simulated data of individual simulated root files (gray triangles for Ruler, pink squares for Timer, and red diamonds for Sizer, $n = 122$ each). Data from individual epidermal files in Col-0 are in green circles [$n = 122$ from day 6, 8, and 10 postgermination (Tables EV1 and EV2)]. Dashed lines are minimum square linear fits. Pearson correlation coefficient ($r$) for each pair of data is indicated. $P$ stands for the $P$-value using standard Pearson correlation test. The epidermis exhibits of the number of cells in the EZ with its length, and also with $1/\ln(r_{EZ})$. The epidermal mature cell length does not have correlation with the inverse of the elongation factor. These features are only reproduced in the Sizer model. For each model, the simulated roots and cells differ in the threshold value for cell elongation termination, the cell elongation rate, the meristematic activity, and the initial cell length, according to a variability inferred from the epidermal Col-0 data (Appendix Fig S4B–F). The parameter values are the same for the three models except for the threshold for cell elongation termination, which is specific of each model and has relative variability of 35% (Ruler), 7% (Timer), and 26% (Sizer) (Table EV3). Parameter values selected such that no statistical significance is found between model (for each model) and epidermal data in any of five phenotypic traits (Wilcoxon rank-sum test, $P$-value > 0.01, Table EV4 and Appendix Fig S4B–F). Continuous lines are theoretical predictions for each model (Appendix Text: Section S1.E). All models have the same dependence on the threshold value for differentiation (blue line), but each model has its own dependence on the spatial profile of cells along the EZ ($r_{EZ}$) (lines in gray for Ruler, black and red for Timer, and red for Sizer). The Timer model depends on the cell elongation dynamics and, in contrast with the Ruler and Sizer models, its relationships are not univocally defined by the spatial profile of cells along the EZ ($r_{EZ}$). For the Timer model, the continuous lines represent the theoretical prediction obtained when either $r_{elong}$ (black line) or $R_{prod}$ (red line) changes.

directly proportional to $1/\ln(r_{EZ})$ (red line in Appendix Fig S9D). The Timer model predicts several distinct outcomes for an EZ with shorter cells (i.e., smaller $r_{EZ}$) depending on whether these elongating cells are shorter because of a smaller relative cell elongation rate or because of a higher production of cells in the meristem (black versus red continuous lines in Appendix Figs S9 and S10). In the Timer model, cells elongate during a fixed time interval and the mature cell length depends only on the cell elongation rate and this time interval. Thus, the Timer model predicts that if cells in the EZ are shorter only because of a slower cell elongation rate, the mature cell length is reduced, and the number of cells in the EZ is not altered, but the length of the EZ decreases because cells are shorter (black lines in Appendix Figs S9 and S10). If shorter cells in the EZ arise only because of faster production of cells in the meristem, then the Timer model predicts that the mature cell length is not changed, but there are more cells elongating, since more cells are created per unit of time, and thus, the EZ length is longer (red line in Appendix Figs S9 and S10). Hence, the Timer model drives similar predictions to the Sizer model when changes in the meristematic activity dominate (Appendix Fig S10). Notice that for the Ruler and Sizer models, the predictions only assumed that cell lengths exhibit an exponential profile as a function of cell position, as seen as a first approximation in real roots (Fig 1C, and Appendix Figs S1 and S2), and does not rely on any specific type of cell elongation dynamics.

Because of these distinct predictions, the simulated roots that emulate the wild type for the Ruler, the Timer, and the Sizer models (Appendix Fig S4B–F) exhibit a variability in the phenotypic traits (mature cell length, number of cells and length of the EZ, and elongation factor in the EZ) that conforms into relationships that depend on the model (Fig 2C–E). This analysis supports that we can give insight into which terminal cell differentiation mechanism underlies primary root growth by evaluating into which relationships wild-type root data are confined.

### The Sizer model is consistent with empirical WT data

The phenotypic traits $l_{max}$, $N_{EZ}$, $L_{EZ}$, and $r_{EZ}$ in the epidermis of each wild-type root, for the total of 122 roots in the stationary regime (days 6, 8, and 10 postgermination, Dataset EV2), were plotted in three selected pairs. We found that the extracted wild-type data followed the three relationships best predicted by the Sizer model (Fig 2C–E): an increase in the EZ length with its number of cells (Fig 2C), no correlation between the mature cell length and the inverse of the elongation factor (Fig 2D) and a proportionality between the number of cells in EZ and $1/\ln(r_{EZ})$ (Fig 2E and Appendix Fig S11). These relationships are also predicted by the Timer model if there is little variability both in the threshold and in the relative cell elongation rate (red line in Fig 2 and Appendix Fig S10). Yet this small variability is not expected (Table EV2).

Single phenotypic traits measured and extracted in the cortex at day 6 postgermination could be all fitted by the three models as well, similarly as showed for the epidermis (Fig 3A, $n = 40$, Tables EV3 and EV4). When evaluating the correlations between the traits, the Sizer model also exhibited a better agreement in the cortex without needing to assume small variability in any parameter, in contrast with the Timer model (Fig 3B–D). Together, the results suggest that the Sizer model describes best the terminal cell differentiation mechanism that ends cell elongation in *Arabidopsis* roots.

### The Sizer model can account for a robust root growth that is proportional to the meristematic activity

We computationally evaluated the significance of the correlation between the number of cells in EZ and $1/\ln(r_{EZ})$ that is found in wild-type data, both in the epidermis and in the cortex. $1/\ln(r_{EZ})$ can be interpreted as the ratio between the meristematic activity and the cell elongation rate in the absence of variability between cells of the same root file (Appendix Text: Section S1A). A disruption of this correlation, while keeping the same statistics for the average number of cells in the EZ and the elongation factor, resulted in an extraordinary increase in the variability of root growth and mature cell length (Fig 4A and B, and Appendix Text: Section S1F). Thus, this relationship reduces the variability of organ growth and ensures that roots have a characteristic growth over time. While the three models predict some degree of correlation between these traits (Figs 2E and 3D), only the Sizer model predicts their proportionality upon large variability of both the relative cell elongation rate and the meristematic activity.

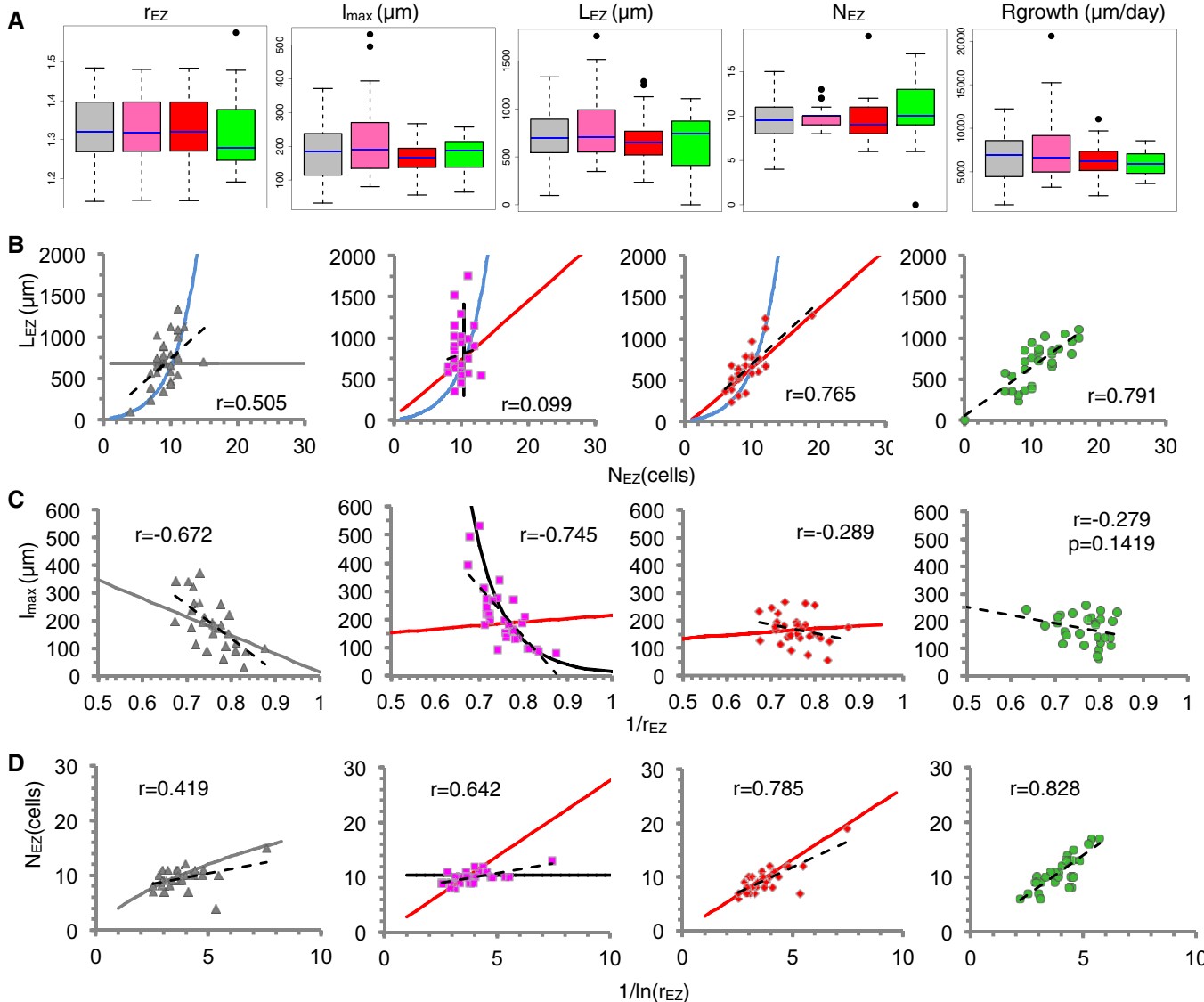

**Figure 3.  Comparison between the predictions of three models for final differentiation with empirical data from the cortex tissue in *Arabidopsis thaliana* roots.**

Simulation results are depicted in gray for Ruler, pink for Timer, and red for Sizer models ($n = 30$ simulated root files, each). Data from individual cortex tissue files in Col-0 are in green [$n = 30$, day 6 postgermination (Tables EV1 and EV2)]. Parameter values for the models (Table EV3) selected such that no statistical significance is found between model (for each model) and cortex data in any of the five phenotypic traits depicted in (A) (Wilcoxon rank-sum test, *P*-value > 0.01, Table EV4). For each model, the simulated roots and cells differ in the threshold value for cell elongation termination, the cell elongation rate, the meristematic activity, and the initial cell length (Table EV3). The parameter values are the same for the three models except for the threshold for cell elongation termination, which is specific of each model and has relative variability of 40% (Ruler), 3% (Timer), and 32% (Sizer) (Table EV3).

A    Boxplots for five phenotypic traits: elongation factor $r_{EZ}$, mature cell length (length of the EZ cell closest to the DZ) $l_{max}$, length of the elongation zone $L_{EZ}$, number of cells in the elongation zone $N_{EZ}$, and root growth rate. For the cortex Col-0 data, the first four phenotypic traits are all measured in the same root files ($n = 30$, day 6 postgermination), while the root growth rate is measured on a different set of roots ($n = 22$, Materials and Methods, also depicted in Appendix Fig S4F). The Ruler and Timer models drive larger variability in the root growth rate. Boxes represent the interquartile range (25th–75th percentiles, with the median indicated by the blue horizontal line) of the distribution. Boxplots were generated using the R function 'boxplot'.

B–D    Relationships between pairs of the phenotypic traits depicted in (A). Panels from left to right: the Ruler, Timer and Sizer models, and cortex Col-0 data. Symbols in the three left-most panels represent simulated data of individual simulated root files (gray triangles for Ruler, pink squares for Timer, and red diamonds for Sizer, $n = 30$ each). Data from individual cortex files in Col-0 are in green circles ($n = 30$). Dashed lines are minimum square linear fits. Pearson correlation coefficient ($r$) for each pair of data is indicated. *P* stands for the *P*-value using standard Pearson correlation test. Continuous lines as in Fig 2C–F but for the corresponding parameter values (Table EV3).

Additionally, the effects on root growth and on the mature cell length driven by changes in the meristematic activity and how these depend on the observed correlation were analyzed computationally (Fig 4C–D and Appendix Text: Section S1F). We found that the speed of stationary root growth increases with the meristematic activity when the correlation holds (Fig 4E). This is because the

average length of differentiated cells is independent of changes in the meristematic activity in this case (Fig 4G). When the correlation is absent, a reduced meristematic activity drives an extremely faster root growth, with very long mature cells (Fig 4F and H).

Together, our computational analysis predicts that the correlation between $N_{EZ}$ and $1/\ln(r_{EZ})$ uncovered in wild-type data is essential to account for reduced root growth with unaltered mature cell length when the meristematic activity decreases. This prediction coincides with the decreased root growth and an unaltered mature cell length reported previously (Chaiwanon & Wang, 2015; Rodriguez *et al*, 2015; Street *et al*, 2016).

## Brassinosteroid signaling at the meristem is sufficient to modulate the threshold for final cell differentiation

We turned the analysis on a mutant with pleiotropic defects in root growth. To this end, we selected the mutant lacking the BRI1 activity, *bri1-116* mutant (Li *et al*, 2001; De Grauwe *et al*, 2005; González-García *et al*, 2011; Hacham *et al*, 2011; Fig 5A–C). BRI1 signaling in the root epidermis is sufficient to drive wild-type root growth (Hacham *et al*, 2011). In *bri1-116*, the reduced meristematic activity arises from both reduced number of cells and reduced mitotic activity at the meristem (González-García *et al*, 2011). The *bri1-116* mutants exhibit the additional phenotype of a reduced mature cell length in both epidermal and cortical tissues (González-García *et al*, 2011; Hacham *et al*, 2011). In addition, a reduced cell elongation rate has been also reported in *bri1-2* mutant roots (Cole *et al*, 2014).

We measured the cell length along epidermal files for a total of $n = 266$ *bri1-116* roots, pulled from day 2 to 10 postgermination to establish the stationary root growth in these mutants (Appendix Fig S12 and Dataset EV1). Our automated framework-based analysis confirmed the reduced meristematic activity, through reduced meristem size and mitotic activity, and the reduced cell elongation rate and mature cell length (Appendix Fig S12, Table EV5 and Dataset EV2). These reductions were found in both the epidermis and cortex. In addition, the data showed that the stationary state is reached earlier than in the wild type (Appendix Fig S12).

To test which elongation termination mechanism is present in the *bri1-116* mutant, we focused on the phenotypic traits "number of cells in EZ" ($N_{EZ}$), "length of EZ" ($L_{EZ}$), and "elongation factor" ($r_{EZ}$) in single roots in the stationary regime ($n = 126$ for epidermis, corresponding to days 6, 8, and 10 postgermination, Dataset EV2) and on whether the three models could reproduce them. As expected (see Introduction), all three models can account for the change observed in each phenotypic trait of *bri1-116* (Appendix Fig S13A and Table EV4). All models involve a reduced meristematic activity and relative cell elongation rate compared to the WT, but only the Ruler and Sizer models have a reduced threshold of differentiation (Table EV3). As done for the WT data, we then assessed whether the relationships between these traits are best described by any of the models (Fig 5D, F, and H). In contrast to the WT, the correlations between these traits in the *bri1-116* mutant exhibit features of both the Timer and Sizer models (Fig 5D–I and Appendix Fig S13B–D). This suggests that cell elongation termination in the *bri1-116* mutant proceeds through a mixed mechanism between the Sizer and the Timer. Therefore, the absence of BRI1 signaling enables the Timer mechanism to take a role, which is not

apparent in the WT. This suggests that BR signaling through BRI1 facilitates the Sizer mechanism to dominate in the WT.

To avoid meristematic defects in this mutant, we expressed BRI1 specifically in the meristematic cells of *bri1-116* mutants using the pRPS5a promoter (Weijers *et al*, 2001). The automated root analysis of stable T4 homozygous plants confirmed that the extracted meristem size corresponds to this domain of dividing cells (Table EV6 and Fig 5C). Noteworthy, these plants show that BR signaling at the meristem is sufficient to drive proper root growth (Fig 5A and B). Furthermore, the quantification of these roots revealed that the expression of BRI1 at the meristem restored several phenotypic features to wild-type values like the meristematic activity, the cell elongation rate, and the mature cell length (Table EV5). The mature cell length in the cortex, but not in the epidermis, did not exhibit correlation with the elongation factor, pinpointing a partial restoration of the Sizer mechanism (Fig 5G). $N_{EZ}$ and $1/\ln r_{EZ}$ exhibited also a linear correlation with a similar slope to the WT, supporting a restoration of the differentiation threshold (Fig 5I). Yet, the relationship between the length and the number of cells of the EZ exhibited a reduced correlation (Fig 5E). Together, our results suggest that BRs are sufficient at the meristem to restore root growth and the mature cell length and that BR signaling is required to make the Sizer mechanism the dominant one.

## The Sizer model is also consistent with data on roots with reduced Gibberellin biosynthesis

The previous results suggest cells can sense their length to stop cell expansion and that the meristem can be relevant to dictate elongation termination in the EZ through BR signaling. We next evaluated the short root phenotype of plants inhibited for the biosynthesis of gibberellin, which have reduced root growth and mature cell lengths like *bri1-116* roots (Ubeda-Tomás *et al*, 2008; Band *et al*, 2012). We analyzed the cortex file in WT roots with chemical inhibition of gibberellin biosynthesis using paclobutrazol (PAC), which drives reduced root growth like gibberellin biosynthesis mutants (Band *et al*, 2012). In agreement with reported results for PAC concentrations 1 and 5 μM (Band *et al*, 2012), we found that these chemically treated roots have a strongly decreased root growth ($n = 63$ for Col-0 control, $n = 40$ for Col-0 + 1 μM PAC, and $n = 25$ for Col-0 + 5 μM PAC) and a reduced cortex mature cell length, this latter being similar at both concentrations of PAC (Appendix Fig S14, $n = 40$ for Col-0, $n = 29$ for Col-0 + 1 μM PAC, and $n = 19$ for Col-0 + 5 μM PAC, all plants at day 6 postgermination, Datasets EV1 and EV2). Our analysis in Fig 4 indicates that the correlation between the number of cells in the EZ ($N_{EZ}$) and the spatial increase in cell length along the root (as measured by $1/\ln(r_{EZ})$) is relevant to account for roots with equal mature cell length, but distinct root growth. The same analysis predicts that roots treated chemically at these two different concentrations will both exhibit such a correlation.

As done with the WT and *bri1-116* mutant, all three models could be fitted to adjust to single phenotypic traits of root growth in the roots grown under the two different concentrations of PAC (Fig 6A for 1 μM PAC and Appendix Fig S16 for 5 μM PAC, Table EV4). All models involved a change in the differentiation threshold, the relative cell elongation rate, and the meristematic activity, compared to the WT (Table EV3). We then evaluated the three relationships between the phenotypic traits of $l_{max}$, $L_{EZ}$, $N_{EZ}$,

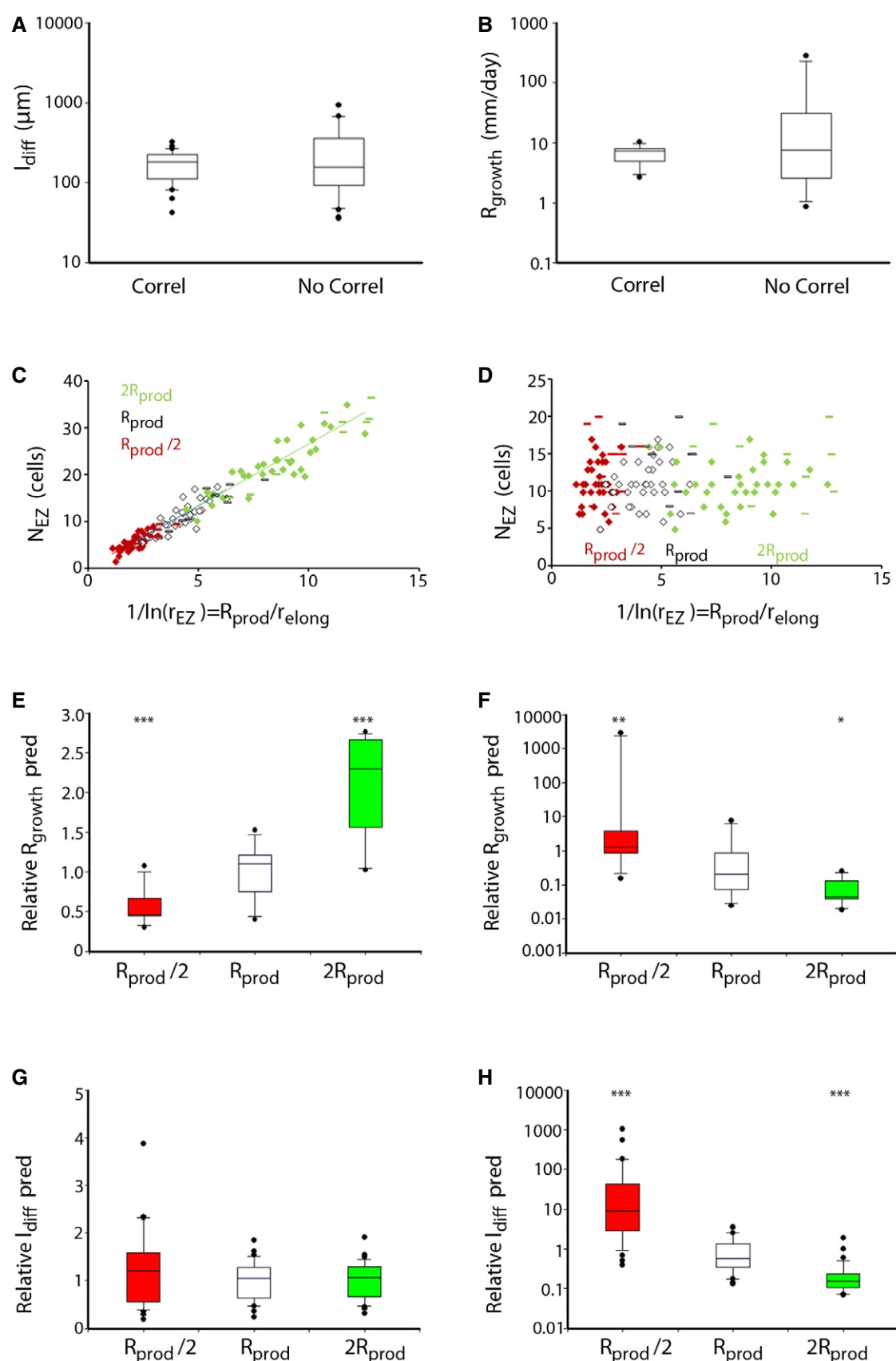

**Figure 4.**

◄

**Figure 4.   Implications of the root meristem–elongation zone correlation.**

A     Theoretical distribution of lengths of mature cells when $N_{EZ}$ correlates with $1/\ln(r_{EZ})$ (as in panel C) and when it does not (as in panel D). In the correlated case, the values extracted for real wild-type plants for $l_{oEZ}$, $r_{EZ}$ and $N_{EZ}$ were used ($n = 122$). For the non-correlated case, the real values of $r_{EZ}$ and $N_{EZ}$ were randomly coupled in pairs and the $l_{oEZ}$ value for the associated $r_{EZ}$ was used.

B     Theoretical distribution of root growth computed when $N_{EZ}$ correlates with $1/\ln(r_{EZ})$ (as in panel C) and when it does not (as in panel D). In the correlated case, the values extracted for real wild-type plants for $R_{prod}$, $l_{oEZ}$, $r_{EZ}$, and $N_{EZ}$ were used ($n = 24$, white symbols in panel C). For the non-correlated case, the real values of $r_{EZ}$ and $N_{EZ}$ were randomly coupled in pairs and the $R_{prod}$, $l_{oEZ}$ values for the associated $r_{EZ}$ were used.

C–D   $N_{EZ}$ versus $1/\ln(r_{EZ})$ in the presence or absence of the meristem–elongation correlation, when $R_{prod}$ is changed twofold.

E–H   Theoretical distribution of mature cell length (E, F) and of root growth (G, H) in the presence (E, G), or absence (F, H) of the correlation, for different values of the meristematic activity ($R_{prod}$) ($n = 35$). In all cases, data were computed using the real experimental values for $1/\ln(r_{EZ})$, $1/(2\ln(r_{EZ}))$, and $2/\ln(r_{EZ})$ and hypothetical values for $N_{EZ}$ chosen to either preserve or not the correlation. ***$P < 0.005$, **$P < 0.01$, *$P < 0.05$. Expressions $l_{diff} = l_{oEZ} r_{EZ}{}^{N_{EZ}}$ and $R_{growth} = R_{prod} l_{diff}$ were used. Relative values were obtained by dividing $l_{diff}$ over the mean at case $R_{prod}$.

Data information: Boxes represent the interquartile range ($25^{th}$–$75^{th}$ percentiles, with the median indicated by the black horizontal line) of the distribution, whiskers extend to the $10^{th}$ and $90^{th}$ percentiles and outliers are represented by black dots. Statistical methods described in Materials and Methods.

and $r_{EZ}$ in these chemically treated plants roots (Fig 6B–D for 1 μM PAC and Appendix Fig S16 for 5 μM PAC, Dataset EV2). As predicted from our computational analysis in Fig 4, $N_{EZ}$ is correlated with $1/\ln r_{EZ}$ in these plants (Fig 6D and Appendix Fig S15). In addition, the three relationships are of the same type as in the WT (Fig 6B–D and Appendix Fig S16), suggesting that the Sizer mechanism drives cell elongation termination in these roots. Comparison of these relationships with those arising from each model confirms that the Sizer model describes best the correlations between these traits (Fig 6B–D and Appendix Fig S16). While the Ruler model is not able to drive such relationships, the Timer model would only drive them when the variability in the threshold time and in the relative cell elongation rate is assumed to be small enough, which is not expected to happen based on our quantification of dynamical traits.

These results indicate that impairment of GA signaling, in contrast with impairment of BR signaling, does not change which is the mechanism that settles cell elongation termination. Our data support the Sizer mechanism as the one taking place when GA biosynthesis is impaired, as in the WT. Yet, GA biosynthesis impinges on cell elongation termination by modifying the threshold length.

# Discussion

In growing organisms, developmental decisions are robust, although biochemical processes and phenotypic traits present an important degree of stochasticity (Oates, 2003; Eldar & Elowitz, 2010; Balázsi *et al*, 2011; Garcia-Ojalvo & Martinez Arias, 2012; Meyer & Roeder, 2014). This is also the case for plant development and growing roots (Roeder *et al*, 2010; Hong *et al*, 2016; Meyer *et al*, 2017). For instance, the mitotic events at the root meristem or the size cells need to reach before dividing, all show variability (Roeder, 2012; Mä *et al*, 2014). This raises the question of how plants regulate size and how they cope with such stochastic cellular behavior in order to generate characteristic cell lengths and organs (Powell & Lenhard, 2012). In this study, we computationally show that the variability of phenotypic traits arising from variability in cell growth parameters is constrained by the developmental mechanisms underlying them. In addition, our results underscore the relevance of analyzing individual roots and the information contained in the variability they exhibit.

Here, we investigated several distinct mechanisms for terminal cell differentiation, the Ruler, Timer, and Sizer mechanisms, which differ in the feature sensed by the cells: either distances, time, or cell lengths, respectively. Through mathematical modeling, we showed that each mechanism can drive unique quantitative relationships between phenotypic traits, especially when the threshold value for differentiation does not vary. We used these differences to suggest the mechanism underlying root growth. Based on the predictions raised by each model in the WT, we propose the Sizer mechanism drives termination of cell expansion in these roots.

The models and methods presented can be applied to any mutant or chemically treated root with stationary root growth, as we showed for roots with altered BR or GA signaling. They can also be used to analyze the stationary root growth of other species with meristematic, elongation, and differentiation zones. Poplar and maize are among the most suitable candidates, especially since new techniques are developed for more robust *in vivo* measurements in these systems (Bizet *et al*, 2014).

In this study, quantitative computational modeling was essential in different aspects. It was necessary to predict quantitative features, such as which relationships between phenotypic traits are established by each mechanism that makes cell to end elongation. It enabled us to validate the inference of dynamical parameters. Moreover, modeling unveiled the relevance of the linear correlation between traits of the MZ and the EZ (i.e., $1/\ln(r_{EZ})$ and number of cells in the EZ) to drive characteristic root growth despite stochastic variability. We found this correlation essential to predict phenotypic changes in agreement with those reported in roots in which the meristematic activity is altered, but not the mature cell length (Chaiwanon & Wang, 2015; Rodriguez *et al*, 2015; Street *et al*, 2016). Similarly, we found it in roots treated with PAC at different concentrations, which are known to exhibit the same mature cell length, but distinct root growth (Band *et al*, 2012). Importantly, the combination of both computational and experimental quantitative approaches enabled to pinpoint the Sizer model as the most plausible one to account for stationary root growth in *Arabidopsis*. The actual mechanism may involve additional complexities together with the Sizer model. For instance, a mechanism enabling but not dictating final differentiation could be coupled to a Sizer mechanism, without altering it. For instance, PLETHORA family form a gradient in the MZ from the QC that is required to enable, but not mediate cell differentiation (Mähönen *et al*, 2014).

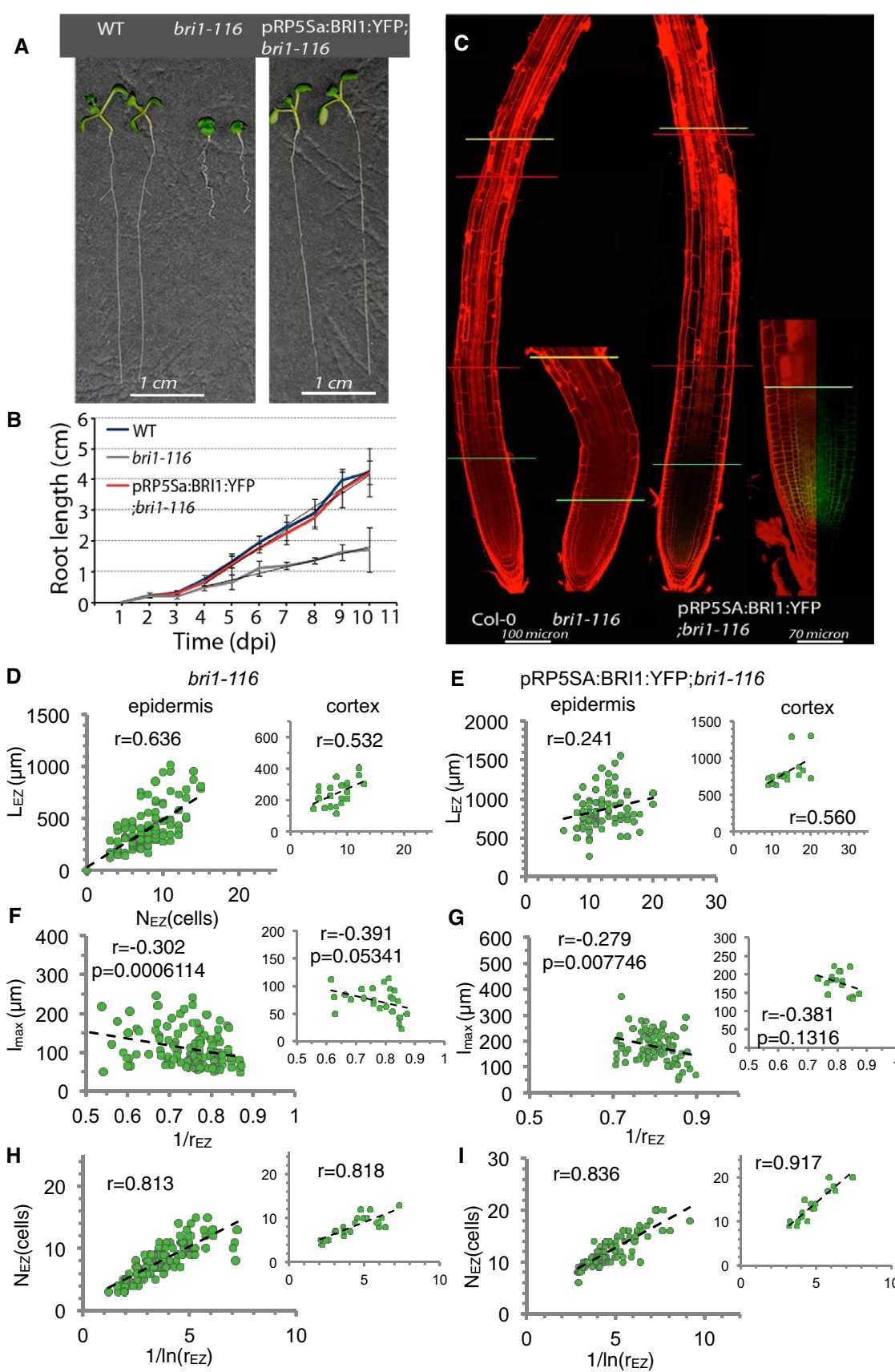

**Figure 5.**

**Figure 5. BRI1 signaling at the dividing cells restores overall root growth.**

A    Eight-day-old seedlings of WT, *bri1-116*, and pRP5Sa:BRI1:YFP;*bri1-116*. Scale bar 1 cm.

B    Root length measurements of 10-day-old *bri1-116*, *bes1-D*, and pRP5SA:BRI:YFP;*bri1-116* seedlings compared to the wild type. Symbols represent the mean of more than 20 plants, from three independent experiments. Error bars show standard deviation. Straight lines represent the linear regression applied to each curve starting with day 4 postgermination. In this way, the root growth rate can be extracted for each phenotype (see Table EV7).

C    Juxtaposed (red lines) confocal images of 8-day-old WT, *bri1-116*, and pRP5SA:BRI:YFP;*bri1-116* roots stained with PI. Green lines label the end of the transition zone, and the yellow lines label the first root hair (end of elongation zone). The inset shows pRP5SA expression domain of an 8-day-old pRP5SA:BRI:YFP;*bri1-116* seedling. Green line labels the end of the meristematic zone, which coincides with the end of pRP5SA expression domain. Scale bars correspond to 100 μm.

D–I    Main relationships as in Fig 2C–E between experimental values of the phenotypic traits used to assess the mechanism of differentiation in *bri1-116* mutant (D, F, H) and pRP5SA:BRI:YFP;*bri1-116* line (E, G, I) for the epidermis (left large plot) and cortex (right inset plot). Each circle denotes data extracted from a single root ($n = 126$ (epidermis) and $n = 25$ (cortex) for *bri1-116* and $n = 90$ (epidermis) and $n = 17$ (cortex) for pRP5SA:BRI:YFP;*bri1-116*, see Table EV1, pooled from 6-, 8-, and 10-day-old seedlings for epidermis and from day 6 for cortex [data in Table EV5)]. In (H, I), the slope of the linear regression for epidermal data is $2.13 \pm 0.04$ for *bri1-116* and $2.53 \pm 0.04$ for pRP5SA:BRI:YFP;*bri1-116*. Pearson correlation coefficient $r$ is indicated and $P$-value ($P$, using standard Pearson correlation test). In (F), *bri1-116* epidermal data do not conform to normal distributions (Spearman correlation test results in (epidermis) $r = -0.29$, $P = 0.00104$ and (cortex) $r = -0.54$, $P = 0.00546$ for *bri1-116*, and in (epidermis) $r = -0.24$, $P = 0.020$ and (cortex) $r = -0.38$, $P = 0.136$ for pRP5SA:BRI:YFP;*bri1-116*).

Recently, the epidermal cells in the *Arabidopsis* shoot stem cell niche were quantified to assess cell size regulation triggering cytokinesis. The quantification showed that expanding cells do not enter this cell-cycle phase once reaching a threshold cell size, or cell size increment or after a time interval. Instead, the data support a mixed scenario between the cell size and size increment paradigms, and not with the Timer-like mechanism (i.e., time interval paradigm). It is interesting that the paradigms of cell size increment and cell size can be thought as both corresponding to sensing the cell size: either the absolute cell size or its increment, respectively. Additionally, it is worth noticing that the Timer-like mechanism is excluded. Indeed, since these cells expand exponentially with a constant relative cell elongation rate (Willis *et al*, 2016), the Timer mechanism could potentially drive much more variability than the other mechanisms yielding less robust outcomes, as we have shown for roots. The large variability associated with the Timer mechanism is well known in the context of the cell cycle, and because of it, the Timer mechanism is not thought to be an appropriate way, on its own, to set stereotyped sizes (Amodeo & Skotheim, 2016). In our context of root growth, the Timer model could drive similar outcomes as the Sizer model only under very restrictive conditions of small variability in the threshold time and the cell elongation rate, suggesting it is a less plausible scenario.

Yet, our results suggest that the Timer mechanism is involved in root growth in the absence of BRI activity. Moreover, the analysis of similar short root phenotypes that are driven by the alteration of a distinct hormone signaling (i.e., GA signaling) shows this role of BRI1 is specific. Hormone signaling gradients can be expected to mediate Ruler, Timer, or Sizer mechanisms depending on how the gradient is formed (Appendix Fig S17). For instance, when diffusion and degradation drive a spatial signaling gradient, the concentration of the signaling molecules depends on their spatial position relative to where they were produced (Crick, 1970). Thus, sensing this signaling gradient can provide positional information and mediate a Ruler mechanism (see Appendix Fig S17 for simulations). In contrast, a signaling gradient formed only by dilution within cells expanding and becoming displaced is dictated by the cell length (Band *et al*, 2012) and thereby can mediate a Sizer mechanism (Appendix Fig S17). For instance, a Sizer mechanism arises when cell elongation terminates once the concentration of this signaling molecule within the cell is below a threshold value (Appendix Fig S17). Computational and mathematical modeling has previously proposed that despite being diffused, gibberellin is synthetized mainly at the meristem and its concentration decays across the EZ mostly through dilution (Band *et al*, 2012). Hence, GA concentration across the EZ is dictated by the cell length and not by the distance from the meristem and therefore can potentially mediate the Sizer mechanism. Then, the GA concentration would be below threshold in shorter cells than in the WT when GA biosynthesis is inhibited (but not completely blocked). Thus, reduced GA biosynthesis at the meristem would drive a reduced mature cell length (besides changes in relative cell elongation rate and meristematic activity) and roots should still exhibit the features of the Sizer mechanism, as we find for roots treated with 1 μM PAC. However, additional reduction in biosynthesis, by higher PAC concentrations, would be expected to drive shorter mature cell lengths (unless additional assumptions are made), in contrast with what is found and suggesting that the GA gradient may not underlie the Sizer mechanism of cell elongation termination in roots. Moreover, recent visualization of the GA gradient in roots challenges the GA gradient itself (Rizza *et al*, 2017). Alternatively, GA signaling could participate in cell length sensing by modulating BR signaling components through their crosstalk. Crosstalk between BRs and GAs can occur at the level of signaling, such that BR signaling components downstream BRI1 receptor (such as BZR) interact with growth repressors DELLA proteins, which are inhibited by GA, and/or at the level of GA biosynthesis (Ross & Quittenden, 2016). Moreover, we show that BRI1 signaling at the meristem is sufficient for root growth. BRI1 expression under another promoter located at the meristem, RCH1, in the *bri1* mutants partially rescued the wild-type phenotype (Hacham *et al*, 2011). In addition, our results suggest that BRI1 signaling is required for the Sizer mechanism to be dominant and that the Timer mechanism plays a role, together with the Sizer, in its absence. A potential scenario can be envisaged to account for these results based on signaling gradients across the EZ formed by dilution in expanding cells and in cell elongation termination below a threshold signaling concentration (Appendix Fig S17). If such a signaling molecule becomes degraded in the absence of BRI1 signaling, and not only diluted by cell expansion, then its concentration within the cells depends on time and on the cell size. Therefore, sensing this molecule for terminating cell elongation would result in a mixed Timer and Sizer mechanism (Appendix Fig S18). Instead, if the molecule is very stable and is not degraded when BRI1 signaling is present, it only mediates the Sizer mechanism since its concentration depends only on the cell size (Appendix Fig S17). Changes in the anisotropic cell growth as well as the effect of temperature on

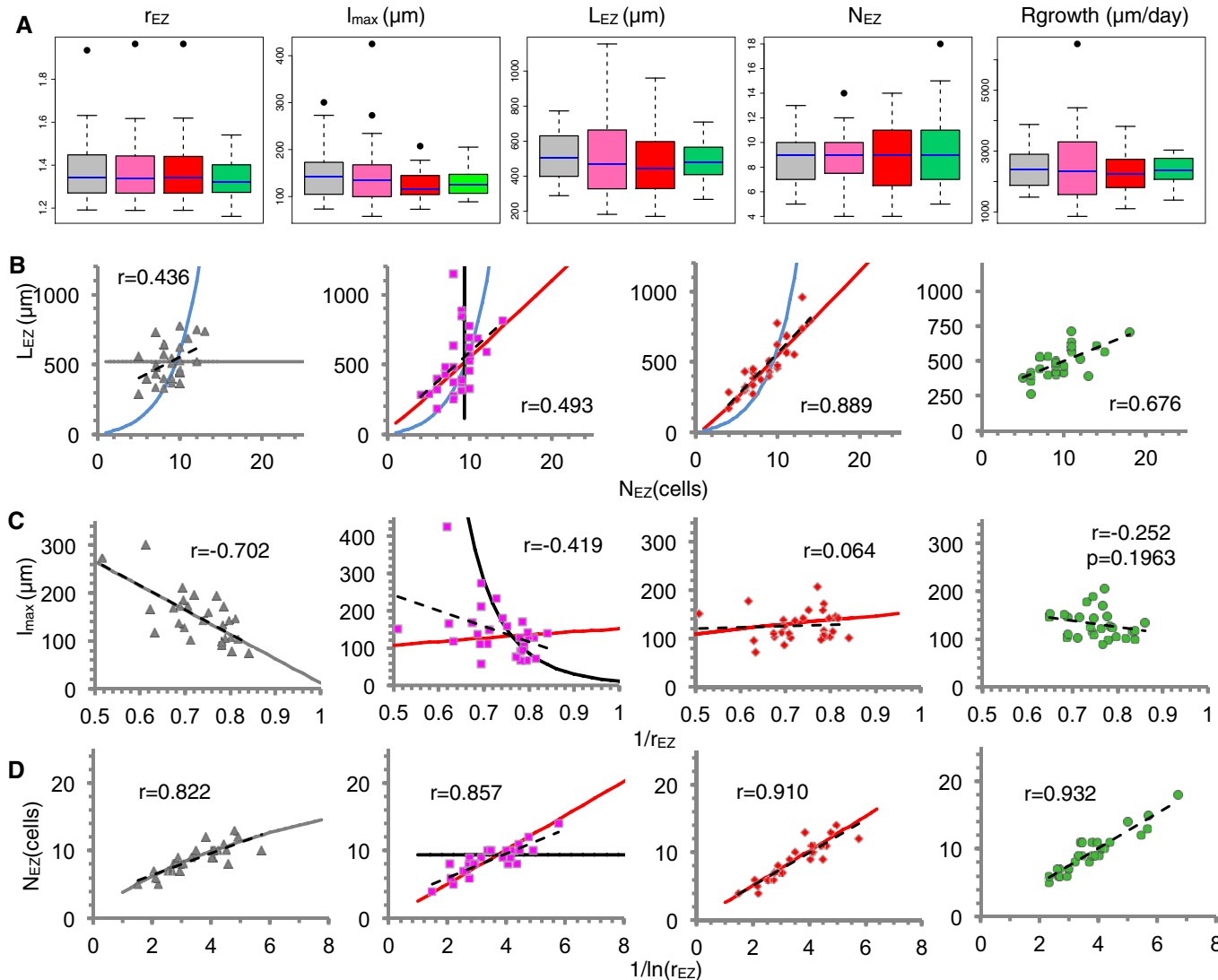

**Figure 6. Comparison between the predictions of three models for final differentiation with empirical data from the cortex tissue in WT *Arabidopsis thaliana* roots grown with 1 µM PAC.**

Simulation results are depicted in gray for Ruler, pink for Timer, and red for Sizer models (*n* = 28 each). Data from cortex tissue in Col-0 grown with 1 µM PAC are in green (*n* = 28, day 6 postgermination). Parameter values for the models (Table EV3) selected such that no statistical significance is found between model (for each model) and cortex data in any of the five phenotypic traits depicted in (A) (Wilcoxon rank-sum test, *P*-value > 0.01, Table EV4). For each model, the simulated roots and cells differ in the threshold value for cell elongation termination, the cell elongation rates, the meristematic activities, and the initial cell length. The parameter values are the same for the three models except for the threshold for cell elongation termination, which has relative variability of 16% (Ruler), 2% (Timer), and 14% (Sizer).

A   Boxplots for phenotypic traits: elongation factor $r_{EZ}$, mature cell length (length of the EZ cell closest to the DZ) $l_{max}$, length of the elongation zone $L_{EZ}$, number of cells in the elongation zone $N_{EZ}$, and root growth rate. For the cortex data, the first four phenotypic traits are all measured in the same root files (*n* = 28, day 6 postgermination), while the root growth rate is measured on a different set of roots (*n* = 40, Materials and Methods). Boxes represent the interquartile range ($25^{th}$–$75^{th}$ percentiles, with the median indicated by the blue horizontal line) of the distribution. Boxplots generated using the R function 'boxplot'.

B–D   Relationships between pairs of the phenotypic traits depicted in (A). Panels from left to right: the Ruler, Timer and Sizer models, and empirical cortex Col-0 data. Symbols in the three left-most panels represent simulated data. Continuous lines are theoretical predictions for each model (Appendix Text: Section S1.E). Dashed lines are minimum square linear fits. Pearson correlation coefficient *r* for each pair of data. *P* stands for the *P*-value using standard Pearson correlation test.

the Timer response could be included and evaluated. Yet, other or additional molecular mechanisms could be driving the sensing of the cell length mediated by BR signaling. For instance, cell sensing in the elongated cell geometries of epidermal and cortex EZ cells could be through intracellular gradients, as proposed in other organisms (see Amodeo & Skotheim, 2016; Marshall *et al*, 2012 for reviews). Because expression of BRI1 only at the developing

protophloem is sufficient for proper root growth (Kang *et al*, 2017), another possibility is the participation of carbon flow. Since BR-mediated cell elongation in *Arabidopsis* roots can also be achieved through endoreduplication (Breuer *et al*, 2007) and cell wall synthesis (Fridman *et al*, 2014), the molecular elements controlling these processes can be also good candidates to underlie how the Sizer mechanism is molecularly executed and contributes to root growth.

# Materials and Methods

## Computational models

Each simulated root (i.e., we only simulated one root file for each simulated root) was labeled by index j (j = 1, 2, …, 121). In each simulated root file j, a cell labeled i was created (i.e., entered the elongation zone) at time $t_{ij}$ with length $l_{0ij}$. This cell elongated along time t with a constant relative cell elongation rate $r_{elong\ ij}$ according to $l_{ij}(t) = l_{0ij} \exp(r_{elong\ ij}(t - t_{ij}))$ until it stopped elongation (or until final time of the whole root file simulation ($t = 10$ days) was reached). For each cell i that stopped elongation, the differentiation zone increased in number of cells by one, and in length by the length of this cell i when it stopped elongating. For $t < 10$ days, cell i stopped elongation according to the differentiation rule specific of each model (Fig 2A and B): In the Ruler model, cell i stopped elongating when its center was located at a distance equal or larger than $L_{0j}$ from the end of the meristem. The distance of cell i from the end of the meristem was measured as the sum of all lengths of all cells being elongating [included half (because of the center location) of that of cell i]; in the Timer model, cell i stopped elongating when it had been elongating for a given time $T_{0j}$ ($t - t_{ij} = T_{0j}$); in the Sizer model, cell i stopped elongating when its length was $l_{diff0\ j}$. The length of cell i when entering the differentiation zone was set as the length the cell had when it stopped elongating. For simplicity, in the Ruler model the lengths and the differentiation rule were only evaluated and applied at those times when a new cell was created and not continuously over time. This was not expected to introduce relevant additional factors since the time period between successive entrance of cells used was very small. In all models, the dynamics started with a single cell and showed a transient regime until a rather steady value of $N_{EZ}$ cells in the EZ is settled (notice that the number of cells in the EZ and the EZ length was not fixed nor constant over time in the stationary regime, but could vary around an average constant value). All data for analysis were extracted only in the stationary regime, neglecting the data from the transient regime. Simulations were done in Matlab, Octave, and fortran.

## Parameter values in the computational models

The dynamics of each cell i in a simulated root file j was characterized by four different parameters. (i) Time at which the cell enters the EZ ($t_{ij}$): We set $t_{ij} = 1/R_{prod\ ij}$. (ii) Length of the cell when entering the *EZ*: $l_{0ij}$. (iii) Relative cell elongation rate ($r_{elong\ ij}$). (iv) Differentiation threshold: The threshold value was set to be specific of each simulated root, being the same for all cells in the same root file. For Ruler: distance $L_{0j}$ from the MZ; for Timer: time $T_{0j}$ spend in the EZ; for Sizer: cell length $l_{diff0j}$. Thus, the dynamics of each cell i within the same simulated root j had parameter values $p_{ij}$ where $p$ denotes any of the four parameters. These values were extracted from the same probability distribution for all cells within the same root file, which was assumed as Gaussian with mean value specific of each simulated root ($p_j$) and equal standard deviation $\sigma$ for all roots ($p_{ij} \in N(p_j, \sigma)$) where $N$ stands for Gaussian distribution). This root specific mean value $p_j$ was extracted from a Gaussian distribution with mean value $p$ and standard deviation $\Delta$ with the same distribution used for all simulated roots ($p_j \in N(p, \Delta)$). The

values of $p$, $\Delta$ were chosen based on the ranges of values obtained from our quantifications of single roots for each tissue, genotype, and hormonal treatment in the stationary regime. If the random value gave a zero or negative value of the parameter, this was discarded and another random value was generated from the same Gaussian distribution. The range of variability between cells of the same simulated root ($\sigma$) was assumed to be smaller than between simulated roots ($\Delta$). The values of $p$, $\Delta$, and $\sigma$ for each parameter, model, and each case analyzed are detailed in Table EV3.

## Plant material and growth conditions

Seeds of WT Col-0, *bri1-116*, and the lines pPR5:BRI1-YFP;*bri1-116* were sterilized and grown as described in González-García *et al* (2011). Seeds were surface-sterilized in 35% sodium hypochlorite, vernalized 72 h at 4°C in darkness, and grown on vertically oriented plates containing 1× Murashige and Skoog salt mixture, 1% sucrose, and 0.8% agar. Plates were incubated at 22°C and 70% humidity under long-day conditions (16-h light/8-h dark). The pRP5:BRI1-YFP;*bri1-116* construct was cloned using recombination Gateway Multisite Cloning system. DNA sequences were amplified from genomic DNA. The purified gene PCR products were placed into the gateway pDONOR221 donor vector by BP reaction mixing 50 fmol of PCR product with 150 ng of the pDONOR221 and 1 μl of BP clonase enzyme diluted up to 5 μl in TE pH 8.0. The same procedure was used for promoter sequences placed in the gateway P4P1R vector. For tagged YFP, a P2RP3 donor vector was used. Recombination LR reaction was performed by mixing the three sequenced pDONOR vectors (10 fmol each one) in a three-component 25 fmol pDEST vector (pB7m34GW) adding 2 μl LR clonase enzyme, diluted up to 8 μl in TE buffer pH 8.0.

For WT Col-0 + PAC data shown in Fig 6 and their WT control in Appendix Figs S14–S16, plants were grown in vertical plates containing half strength Murashige and Skoog (MS) medium with vitamins and no sucrose supplements (0.5XMS-) in long-day conditions. For PAC treatments, the compound was dissolved in acetone was added and dissolved in the media to a final concentration of 1 μM or 5 μM.

## Root measurements

Approach 1: Plates containing the growing plants were scanned every 24 h for 10 days, and the root length was measured with ImageJ software (http://rsb.info.nih.gov/ij/) and averaged between all plants for each day. An average root growth rate was extracted from the slope of a least square linear fitting on the average root length (averaged over the n roots). Confocal images were taken at days 1, 2, 3, 4, 5, 6, 8, and 10, on another set of roots allowing the measurement and counting of the cells along single epidermal cell files. Measurements were carried on at least 20 plants for each post-germination day, resulting from more than three independent experiments (see Table EV1). Approach 2: The root length of the same plant was tracked each day from day 4 (starting with this time point root growth becomes linear) to day 8 postgermination (at this time point the meristem size is stationary). At day 8, the plant was transferred to confocal microscopy and the root was imaged. In this way, it was possible to assign a root growth rate to each root by performing a least square linear fitting on each root. Results of both

approaches are shown in Table EV2. Number of roots analyzed are detailed in Table EV1. For PAC treatments and their Col-0 control, the root was imaged for measuring cortex cell lengths at day 6 post-germination. The root growth was extracted by linear fittings on root lengths measured each day from day 4 to day 10, both included, postgermination on another set of roots.

### Confocal microscopy

Different roots were visualized with a FV 1000 confocal microscope (Olympus, Tokyo, Japan) at days 1–6, 8, and 10 after germination in Approach 1 and at day 8 in Approach 2. All roots from day 3 onward were imaged live, being counter-stained with propidium iodide (PI) as described previously in González-García *et al* (2011). Plants from days 1 and 2 postgermination were imaged fixed, by staining using a modified pseudo-Schiff propidium iodide (mPS-PI) staining technique (Truernit & Sauer, 1995). Epidermal and cortical cells were measured individually in each plant along a single file and each analyzed time point using ImageJ software (http://rsb.info.nih.gov/ij/). For epidermal cells, the last cell before the hair cell was the one measured as $l_{max}$. Within the meristem, the hair and non-hair cells could be easily identified based on their morphology. In each root, the focal plane was adjusted to capture the same file of epidermal cells, to ensure consistency between individual plant measurements. Once identified within the meristem, the hair epidermal cell file was followed into the elongation zone. Mature cells for cortical layer were considered the ones at the same longitudinal position as the root hair and xylem differentiation (Ubeda-Tomás *et al*, 2009; Band *et al*, 2012; Mähönen *et al*, 2014).

### Quantitative analysis of individual plant roots

An Octave/Matlab routine was developed to extract several parameters ($r_{MZ}$, $r_{EZ}$, $l_{0MZ}$, $l_{0EZ}$, $N_{MZ}$, $N_{EZ}$, $l_{max}$) from each plant individually (Appendix Fig S3A). The procedure is detailed in Appendix Text: Section S1.B, and the program code is in Appendix Text: Section S3.A. The validation of the fittings is presented in Appendix Fig S3B–F and Appendix Text: Section S1.C. This automated method defined the MZ. Elongation zone size was given by the number of elongated cells, counted along the root from the last cell in the MZ (not included) until the first mature cell in the DZ (appearance of the hair bulge) (not included). Postmeasurement analysis was done in Excel (Microsoft Office), MATLAB R2009b (The MathWorks Inc., Natick, MA, 2000), Octave version 4.2.1, SigmaPlot 11.0 and R. Those roots where no elongation zone was extracted (1 out of 122 for the epidermis and 1 out of 60 for cortex in Col-0 and 1 out of 126 in *bri1-116*) were considered to have $N_{EZ} = 0$ and $L_{EZ} = 0$, and undefined elongation factor and mature cell length. Therefore, all plots involving either the elongation factor or the mature cell length did not have the data from these roots included.

### Statistical methods

Statistical analyses were done with R and SigmaPlot 11.0. In Fig 4, comparisons between data were performed using One Way ANNOVA (when data was normally distributed) and Kruskal-Wallis One Way Analysis of Variance on Ranks (when data was not normally distributed).

### Extraction of dynamical information

Through the quantitative analysis of individual plant roots and from root length measurements over time, an estimation of dynamical parameters such as relative cell elongation rates, meristematic activity rate, and time spent in the EZ was obtained (see description in Appendix Text: Section S1.A and formulae in Table EV2 both for Approach 1 and Approach 2). This procedure is similar to the used in Cole *et al* (2014), being the differences detailed in Appendix Text (Section S2.B). The results are shown in Tables EV2 and EV5 and discussed in Appendix Text (Section S2.B) at the light of previous measurements (Beemster & Baskin, 1998; Fiorani & Beemster, 2006; Band *et al*, 2012; Cole *et al*, 2014). The validation of the procedure was done on the data extracted from simulated roots (see Appendix Text: Section S1.D) and is shown in Appendix Figs S5–S7 for each model.

### Data availability

Two Datasets are provided as Expanded View, each containing several files. Dataset EV1 contains the cell lengths measured along the meristem and the EZ for all roots analyzed. Dataset EV2 contains the results of the automated fitting performed on the data in Datasets EV1.

**Expanded View** for this article is available online.

### Acknowledgements

M.I. acknowledges support from the Ministerio de Economía y Competitividad (Spain) and Fondo Europeo de Desarrollo Regional FEDER (EU) through grant FIS2015-66503-C3-3-P (MINECO/FEDER), from MINECO through FIS2012-37655-C02-02 and FIS2009-13360-C03-01, and from the Generalitat de Catalunya through Grup de Recerca Consolidat 2014 SGR 878. A.I.C.-D. is a recipient of BIO2016-78955 and BIO2013-43873 grants from the Spanish Ministry of Economy and Competitiveness (MINECO), of an ERC Consolidator Grant (ERC-2015-CoG—683163) from the European Research Council, and has funding from Developing an European American NGS Network (DEANN) from FP7-PEOPLE-2013-IRSES (2015–2017). M.G.-G. was the recipient of a postdoctoral contract from BIO2010-16673 grant from the Spanish Ministry of Economy and Competitiveness, J.V.B was funded by FI PhD fellowship from Generalitat de Catalunya, and I.P was funded by a JAE-CSIC PhD fellowship in A. I.C.-D. laboratory. A.P.-R. is a recipient of a PhD fellowship from the "Severo Ochoa Programme for Centers of Excellence in R&D" 2016–2019 (SEV-2015-0533)" in A.I.C-D. laboratory. M.I. acknowledges Andreu Alibés for technical assistance.

### Author contributions

AIC-D and MI conceived the project. IP, JV-B, and AP-R performed the experiments. IP and MI performed the computational analyses. IP, JV-B, M-PG-G, AIC-D, and MI designed the research and critically assessed the data. IP, AIC-D, and MI wrote the manuscript. IP, JV-B, AP-R, M-PG-G, and AIC-D revised the manuscript.

### Conflict of interest

The authors declare that they have no conflict of interest.

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
