## [Review Process File · Molecular Systems Biology]

A Sizer model for cell differentiation in *Arabidopsis thaliana* root growth

Irina Pavelescu, Josep Vilarrasa-Blasi, Ainoa Planas-Riverola, Mary-Paz González-García and Ana I. Caño-Delgado & Marta Ibañes

Review timeline:	Submission date:	12 April 2017
	Editorial Decision:	12 June 2017
	Revision received:	9 October 2017
	Editorial Decision:	7 November 2017
	Revision received:	21 November 2017
	Accepted:	27 November 2017

Editor: Maria Polychronidou

Transaction Report:

1st Editorial Decision

12 June 2017

Thank you again for submitting your work to Molecular Systems Biology. We have now heard back from two of the three referees who had initially agreed to evaluate your study. Reviewer #3 unfortunately never returned a report after several reminders so I additionally invited a fourth reviewer. As you will see below, the three reviewers appreciate that the presented findings seem interesting. However, they raise a series of concerns, which we would ask you to address in a revision.

Without repeating all the points listed below, one of the more fundamental issues raised by both reviewers refers to the need to include further experimental analyses (e.g. using GA and auxin mutants) that could provide some level of mechanistic insight into the proposed Sizer model. Of course all other issues listed by the referees would also need to be addressed.

REVIEWER REPORTS

Reviewer #1:

This is an interesting article on how cell size is regulated during cell differentiation in the root. Combining careful quantitative analysis with computational modelling, the authors provide evidence that final cell size is regulated in a cell autonomous manner by a mechanism that is able to sense cell dimensions.

The manuscript clearly demonstrates how rigorous quantitative biology can provide novel insights, in particular when combined with modelling approaches to help in the interpretation of data where

behaviour in time and space has to be analyzed. The observation that brassinosteroid signaling at the meristem is sufficient to control cell size regulation is certainly of interest. However, in spite of an overall positive impression I have a number of remarks and questions that would need attention.

- The finding that BR in the meristem alone can restore final cell length could imply that BR helps to put in place a sensor for plant size. However, this finding is also in agreement with a Ruler mechanism (e.g. a BR induced or dependent gradient). This ambiguity is underlined in the conclusion where the authors discuss the potential role of gradients (Plethora, Gibberellin). I guess a Sizer mechanism based on the dilution of GA as suggested in the discussion, is equivalent to a gradient based Ruler mechanism (note that GA could in principle also diffuse from cell to cell). I therefore fail to see why they put forward the Sizer hypothesis.

In this context, they might find it useful to look at a recent paper by Willis et al who showed that cell division at the shoot meristem is not determined at a critical size, nor at a particular time interval. The results showed a cell autonomous and intermediate behaviour. Although Ockham's razor would in principle favor a single mechanism, the combination of several processes in parallel can provide robustness (e.g. the role of both cell-cell signalling and division plane orientation in efficiently setting up differentiation patterns in the root, different signaling pathways involved in organ positioning etc.).

- Linked to the previous point, the molecular mechanisms behind a Sizer model remain obscure. The observation that BR in the meristem alone can rescue the cell elongation phenotype is very interesting, but at this stage it is not clear how signaling, more in line with a Ruler mechanism (see also above), would lead to a cell autonomous Sizer mechanism. The authors invoke other hormones, in particular gibberellins. It is strange that this has not at all been exploited experimentally in this study. GA synthesis mutants are available for example. Auxin related mutants might also be perturbed in final cell size.

- The differences in outputs of the three models are difficult to interpret. Why is it not possible to tune the parameters in such a manner that all of them fit the observations? Signaling gradients can have different shapes and operate on changing distances (e.g. depending on variable wall thickness) for example. The authors should give better intuitive explanations for this, in particular for the non specialist readers.

- Minor points :

- the authors oppose dividing and growing cells. It should be noted, that both are growing following the same principles.

- On page 14 the authors conclude that their « results suggest that BR signaling controls the mature cell length ». If final cell length in *br1* mutants is shorter, BR must control final cell size. No simulations are needed for this. In addition, I am not sure till what extent this effect of BR on final cell size can be uncoupled from the effect on cell elongation and meristematic activity as is suggested.

- Apparently the quantitative data on root growth in the EZ are mainly based on fixed tissues. This makes interpretation of the quantitative data more difficult and time consuming. Why is this?

Reviewer #2:

The control of organ size represents a fascinating question for many researchers studying multicellular organisms. The authors focus on a plant model system, the *Arabidopsis* primary root, given the well-established resources and experimental tractability to measure many of the cell scale parameters in their models. Root cells undergo 6-7 rounds of doubling their size and then dividing in the meristem, then after exiting the meristematic zone cells start to rapidly expand in the elongation zone (up to 30 times in the case of non-hair epidermal cells)* before finally ceasing their expansion as they enter a differentiation zone.

The authors initially develop a model that captures the above dynamics as cells transition through

the meristem and elongation zones. Overall, they employ a series of logical assumptions and/or experimentally derived parameters*. Several useful emergent properties are noted including the ability to predict the boundary between meristem and elongation zones.

*The authors state cell length increases of up to 10 times in the manuscript. They measure epidermal cell lengths, but it is not clear whether they are monitoring hair cells or non-hair cells that form distinct files (which differ up to 2 fold in their relative lengths). Both points need to be clarified in the text.

Next, the authors investigated several distinct mechanisms that may control the cessation of cell elongation. They term these contrasting models, Ruler, Timer and Sizer mechanisms, where cells either sense either distances, time or cell lengths respectively as they transit the elongation zone. The authors elegantly describe how, employing mathematical modelling, each mechanism exhibits contrasting root cell growth behaviours. By comparing their modelling predictions to data obtained from wildtype, they report that only the Sizer based model exhibits several core growth features and performs robustly.

Finally, they examine whether the models can capture the impact of perturbing root growth employing mutants defective in perception to the growth promoting hormone Brassinosteroid (BR) or transgenic lines that rescue the BR mutant growth defects* Once again, the sizer model measures up best.

*The authors employ a meristem expressed promoter to drive the expression on the BR receptor BRI1 to rescue the *bri1* mutant.

Surprisingly, the elongation zone defect in *bri1* can also be rescued. This may be related to the recent observation by the Hardke lab that expressing BRI1 in JUST the proto/phloem cells can rescue the *bri1* growth defect (whereas expressing BRI1 in the epidermis only partially rescues it). Could this reflect that BRI1 is controlling organ growth via regulating C flow into the root apex to drive cell expansion?

Given the findings reported by Band et al (2012) where GA is proposed to cause cessation of root cell expansion via a dilution based mechanism (which could provide an elegant basis for the Sizer model), I was surprised that the authors elected to study the role of BR rather than GA? Have they looked at GA mutants or analysed published datasets of these mutants (e.g. Ubeda-Tomas et al, 2009, Current Biology)? This would add a great deal to the likely molecular mechanism underpinning the modelling findings in this manuscript.

In summary, this represents a fascinating manuscript that synthesises modelling simulations and experimental results as expected for journal such as MSB. Their mechanistic findings are also very interesting and concur well with the findings reported by Band and co-authors in 2012.

Reviewer #4

Manuscript by Pavelescu et al, is a combined experimental and theoretical study directed for understanding how plant root zonation is established and maintained. Authors specifically focused on how root cells make a decision to differentiate and how this decision is executed in the spatio-temporal context. A very strong point of this study is an integration of quantitative measurements with the theoretical models. Three mechanisms are proposed so called "ruler", "timer" and "sizer" to explain transition from elongation to differentiation with the latter being more fit to the empirical data. Technical aspects of computer modeling are well defined and justified. However, I am not convinced that this study provides novel enough insights in our understanding of root zonation pattern.

- 1) In my view, all three mechanisms may represent the same concept if one assume that threshold for distance (in ruler model) and timing (in timer model) are not fixed but dynamically changing during root growth as one would expect. In this scenario all three models could be fit to the experimental observations.
- 2) Also choice of thresholds for timer and ruler mechanisms is arbitrary and predictions strongly

depends on this choice. Would authors provide more comprehensive analysis of average THR in ruler and timer models on the model outcome?

3) Page 13 paragraph 2, I had hard time to compare data from *br1-116* mutant and model prediction it is mentioned in the text it fits Sizer model but no actual comparison is made. Also not mechanism is presented to explain how this mutant contributes to reduced size of differentiated cells.

1st Revision - authors' response

9 October 2017

POINT BY POINT RESPONSE:

Reviewer #1:

This is an interesting article on how cell size is regulated during cell differentiation in the root. Combining careful quantitative analysis with computational modelling, the authors provide evidence that final cell size is regulated in a cell autonomous manner by a mechanism that is able to sense cell dimensions.

The manuscript clearly demonstrates how rigorous quantitative biology can provide novel insights, in particular when combined with modelling approaches to help in the interpretation of data where behaviour in time and space has to be analyzed. The observation that brassinosteroid signaling at the meristem is sufficient to control cell size regulation is certainly of interest. However, in spite of an overall positive impression I have a number of remarks and questions that would need attention.

OUR RESPONSE: We acknowledge the encouraging comments and appreciation of our work. As shown below, we followed your suggestions, which drove significant improvements to our revised manuscript.

- The finding that BR in the meristem alone can restore final cell length could imply that BR helps to put in place a sensor for plant size. However, this finding is also in agreement with a Ruler mechanism (e.g. a BR induced or dependent gradient). This ambiguity is underlined in the conclusion where the authors discuss the potential role of gradients (Plethora, Gibberellin). I guess a Sizer mechanism based on the dilution of GA as suggested in the discussion, is equivalent to a gradient based Ruler mechanism (note that GA could can in principle also diffuse from cell to cell). I therefore fail to see why they put forward the Sizer hypothesis.

In this context, they might find it useful to look at a recent paper by Willis et al who showed that cell division at the shoot meristem is not determined at a critical size, nor at a particular time interval. The results showed a cell autonomous and intermediate behaviour. Although Ockham's razor would in principle favor a single mechanism, the combination of several processes in parallel can provide robustness (e.g. the role of both cell-cell signalling and division plane orientation in efficiently setting up differentiation patterns in the root, different signaling pathways involved in organ positioning etc.).

OUR RESPONSE: We agree with the reviewer that this point deserved further clarification. As indicated by the reviewer, a signaling gradient could drive a sensor for tissue distances and hence a Ruler mechanism, and not necessarily a sensor for cell lengths (Sizer mechanism). We have now clarified that the mechanism behind the signaling gradient formation will dictate whether this gradient mediates a Ruler or a Sizer or a Timer (or a mix). If the gradient is formed mainly by diffusion, the concentration of the molecule is dictated by the distance from the source of biosynthesis. Therefore, this gradient mediates a Ruler mechanism. In contrast, if the gradient is mainly formed because of dilution in the expanding cells, then the concentration is dictated by the cell length and the gradient mediates a Sizer mechanism. We now exemplify this with new computational results in Appendix Figure S17. This is explained in the Discussion (lines 515-561).

Band et al PNAS 2012 proposed that Gibberellin forms a gradient along the EZ in which both dilution and diffusion takes place. Yet, they showed that diffusion is expected to be negligible and dilution to be the dominant factor. Thus this gradient can become the mediator of a Sizer mechanism but not a Ruler one. Recent data challenge however this type of gradient (Rizza et al, Nature Plants 2017). This is also commented in the Discussion (lines 523-544).

We have also brought to the discussion the Willis et al., paper (lines 497-514). In addition, following their notation we have renamed the cell elongation rate as the relative cell elongation rate.

- Linked to the previous point, the molecular mechanisms behind a Sizer model remain obscure. The observation that BR in the meristem alone can rescue the cell elongation phenotype is very interesting, but at this stage it is not clear how signaling, more in line with a Ruler mechanism (see also above), would lead to a cell autonomous Sizer mechanism. The authors invoke other hormones, in particular gibberellins. It is strange that this has not at all been exploited experimentally in this study. GA synthesis mutants are available for example. Auxin related mutants might also be perturbed in final cell size.

OUR RESPONSE: We agree that introducing the results of an additional hormone will further contextualize our results. Gibberellins are a great candidate, since it has been previously proposed that their dilution and EZ gradient is relevant for cell elongation termination in roots (Band et al, PNAS 2012), whereas auxin is thought to enable but not to mediate cell differentiation (Mahönen et al, Nature 2014). We have now evaluated wild type (WT) roots chemically inhibited with Paclobutrazol (PAC) for the biosynthesis of gibberellins. These chemically inhibited roots are known to exhibit the same phenotype as gibberellin biosynthesis mutants (Band et al, PNAS 2012). In addition, according to reported data on cortex files in Band et al, PNAS 2012, the WT roots treated with PAC at concentrations 1 μ M and 5 μ M enable to analyse roots with distinct average root growth but the same average mature cortex cell length. This is a scenario we evaluated only computationally in our original manuscript in Fig.3 and which we can now address empirically with these WT plants treated with PAC. Our computational results predicted the existence of a correlation between the number of cells in the EZ and $1/\ln(\text{rez})$ for plants exhibiting these phenotypes.

We have quantified cortex files in single roots treated with these two concentrations of PAC as well as in the WT, following the quantification procedures reported in our original manuscript. Plant materials have been included in Methods (lines 631-635).

The new results are shown in new Fig. 6, and new Appendix Figures S14, S15 and S16. Data in Appendix Fig S14 confirm the reduced root growth and mature cell length previously reported (Band et al, PNAS 2012). We checked these plants follow the same type of exponential profile of cell lengths along the EZ as the WT and performed their automated fitting, as shown in Appendix Fig. S15. Data in Fig.6 and in Appendix Fig. S16 show these roots (for concentrations 1 μ M PAC and 5 μ M PAC, respectively) exhibit the correlation between NEZ and $1/\ln r_{EZ}$ we predicted computationally.

For completeness, we have also performed numerical simulations for the Ruler, Timer and Sizer models to mimic the WT cortex, as well as that treated with 1 μ M PAC and with 5 μ M PAC. These data are shown in Fig. 3 (WT cortex), Fig 6 (WT grown in 1 μ M PAC) and in Appendix Fig S16 (WT grown in 5 μ M PAC). The results for the Sizer model are in better agreement with those coming from cortex tissues in WT roots and in WT roots treated with PAC.

These Gibberellin data are presented and discussed in a novel final section of Results entitled "The Sizer model is also consistent with the reduction of Gibberellin biosynthesis" (lines 414-443). In the Abstract, Introduction and the Discussion these new results are also mentioned.

- The differences in outputs of the three models are difficult to interpret. Why is it not possible to tune the parameters in such a manner that all of them fit the observations ? Signaling gradients can have different shapes and operate on changing distances (e.g. depending on variable wall thickness) for example. The authors should give better intuitive explanations for this, in particular for the non specialist readers.

OUR RESPONSE: We agree with the reviewer that a better explanation was needed. We have now extensively computationally searched for parameter values, within the ranges of values we have observed in plants, that could be consistent with the empirical WT data on root growth rate and length of mature cells. We also evaluated consistency with three additional

phenotypic traits: 1. number of cells in the EZ, 2. length of EZ and 5. elongation factor. Now we show it is possible to tune the model parameters in such a way that all three models drive for each one of these traits values that can not be discarded based on individual statistical significance (Wilcoxon rank sum test between empirical trait and the corresponding one obtained for each model (for $n=1000$ and for n =number of roots), for each trait, drives p -value >0.01 ; The results of these tests are shown in new Table EV4). Therefore, all three models can drive a rather good picture of root growth. The same has been done also for the cortex cells in Col-0 (new Fig. 3), for *bri1-116* epidermis (new Appendix Fig. S13) and for cortex cells of PAC treated plants at the two different concentrations analyzed (Fig. 6, Appendix Fig. S16), together with the root growth of these plants. The new results show that all three (Ruler, Timer and Sizer) models can drive outcomes for each trait that are consistent with the measured and extracted data from these roots (WT roots, mutants and chemically treated roots).

However, as we indicated in our original manuscript, each model predicts distinct relationships between the traits. These differences between models are used to compare them with the data measured and extracted from roots (new panels in Fig.2 for WT (epidermis), new Fig.3 for WT (cortex), new Fig. S13 for *bri1-116* mutant (epidermis), new Fig.6 and new Appendix Fig. S17 for PAC treated roots (cortex)). For this comparison, we have added Pearson correlation coefficient coming from the computational models and from roots and the p -value of significance for the root data with small correlations. Our results indicate that despite all models can drive outcomes at the level of single traits that are consistent with the empirical data, the Sizer model is the one that describes best the relationships between these traits in the WT and in plants with Gibberellin biosynthesis inhibited. In contrast, the *bri1-116* mutant resembles both the Sizer and the Timer models, suggesting that BRI1-mediated signaling participates promoting the Sizer mechanism.

The Results sections "The three models can give distinct quantitative predictions" (lines 238-316), "The Sizer model is consistent with empirical WT data" (lines 318-336) and "Brassinosteroid signaling at the meristem is sufficient to modulate the threshold for final cell differentiation" (lines 366-412) and Discussion (lines 518-554) have been partially rewritten to present all these new analyses and results. In the Result section "The Sizer model can account for a robust root growth that is proportional to the meristematic activity", we have indicated that the three models drive some level of correlation, but proportionality is intrinsic to the Sizer model, to substitute that the correlation is intrinsic to the Sizer model (lines 348-351).

- Minor points :

- the authors oppose dividing and growing cells. It should be noted, that both are growing following the same principles.

OUR RESPONSE: Yes, this has been now clarified in the Introduction (in line 11 it is explicitly indicated. It has also been rephrased in other parts of the introduction for clarity).

- On page 14 the authors conclude that their « results suggest that BR signaling controls the mature cell length ». If final cell length in *bri1* mutants is shorter, BR must control final cell size No simulations are needed for this. In addition, I am not sure till what extent this effect of BR on final cell size can be uncoupled from the effect on cell elongation and meristematic activity as is suggested.

OUR RESPONSE: We have clarified this in the new Introduction (lines 81-105, lines 119-132), indicating that the change in both cell elongation rate and meristematic activity is not sufficient to drive the reduced mature cell length. In addition, since our new results suggest that *bri1-116* mutant executes a Timer mechanism, we have focused on the description of these new results and role (see previous response). We also stress the difference between *bri1-116* roots and WT roots grown with PAC: both exhibit similar shortening of the roots and mature cell length and the parameters used to model them are rather similar (see new Table EV3). Yet, comparison of the models with the measured and extracted data, suggest different roles of each hormone signaling (Abstract; Discussion, lines 515-551). In addition, in the Discussion we propose one potential mechanism by which BRI1 could drive the cell elongation

termination through Sizer mechanism in the WT, resembling also a Timer mechanism in its absence (lines 548-555). New computational data are provided to clarify this potential scenario (Appendix Figs. S7 and S18). This scenario is based on a molecule that is degraded and becomes diluted by cell expansion over time and which controls the termination of cell elongation. In this hypothetical scenario, the BRI1 would help to stabilize the molecule.

- Apparently the quantitative data on root growth in the EZ are mainly based on fixed tissues. This makes interpretation of the quantitative data more difficult and time consuming. Why is this?

OUR RESPONSE: The quantitative data on root growth presented in this study is based on measurements performed on live plants, stained with Propidium Iodine. We have not used fixed tissues. We only used fixed tissues (modified pseudo-Schiff propidium iodine (mPS-PI) staining) to analyse the *bril-116* mutant plants at days 1 and 2 pg, because the roots were very small. This is indicated in Methods (lines 656-663).

Reviewer #2:

The control of organ size represents a fascinating question for many researchers studying multicellular organisms. The authors focus on a plant model system, the Arabidopsis primary root, given the well-established resources and experimental tractability to measure many of the cell scale parameters in their models. Roots cells undergo 6-7 rounds of doubling their size and then dividing in the meristem, then after exiting the meristematic zone cells start to rapidly expand in the elongation zone (up to 30 times in the case of non-hair epidermal cells)* before finally ceasing their expansion as they enter a differentiation zone.

The authors initially develop a model that captures the above dynamics as cells transition through the meristem and elongation zones. Overall, they employ a series of logical assumptions and/or experimentally derived parameters*. Several useful emergent properties are noted including the ability to predict the boundary between meristem and elongation zones.

*The authors state cell length increases of up to 10 times in the manuscript. They measure epidermal cell lengths, but it is not clear whether they are monitoring hair cells or non-hair cells that form distinct files (which differ up to 2 fold in their relative lengths). Both points need to be clarified in the text.

OUR RESPONSE: We acknowledge the positive evaluation of our work. The results presented in this study for epidermal cells were obtained by monitoring hair cells. Within the meristem, the hair and non-hair cells could be easily identified based on their morphology. In each root, the focal plane was adjusted to capture the same file of epidermal cells, to ensure consistency between individual plant measurements. Once identified within the meristem, the hair epidermal cell file was followed into the elongation zone. This has been now clarified in Methods (lines 663-669).

Next, the authors investigated several distinct mechanisms that may control the cessation of cell elongation. They term these contrasting models, Ruler, Timer and Sizer mechanisms, where cells either sense either distances, time or cell lengths respectively as they transit the elongation zone. The authors elegantly describe how, employing mathematical modelling, each mechanism exhibits contrasting root cell growth behaviours. By comparing their modelling predictions to data obtained from wildtype, they report that only the Sizer based model exhibits several core growth features and performs robustly.

Finally, they examine whether the models can capture the impact of perturbing root growth employing mutants defective in perception to the growth promoting hormone Brassinosteroid (BR) or transgenic lines that rescue the BR mutant growth defects* Once again, the sizer model measures up best.

*The authors employ a meristem expressed promoter to drive the expression on the BR receptor

BRI1 to rescue the bri1 mutant.

Surprisingly, the elongation zone defect in bri1 can also be rescued. This may be related to the recent observation by the Hardke lab that expressing BRI1 in JUST the proto/phloem cells can rescue the bri1 growth defect (whereas expressing BRI1 in the epidermis only partially rescues it). Could this reflect that BRI1 is controlling organ growth via regulating C flow into the root apex to drive cell expansion?

OUR RESPONSE: We appreciate this interesting suggestion, which we have now added at the end of the Discussion section (lines 555-557).

Given the findings reported by Band et al (2012) where GA is proposed to cause cessation of root cell expansion via a dilution based mechanism (which could provide an elegant basis for the Sizer model), I was surprised that the authors elected to study the role of BR rather than GA? Have they looked at GA mutants or analysed published datasets of these mutants (e.g. Ubeda-Tomas et al, 2009, Current Biology)? This would add a great deal to the likely molecular mechanism underpinning the modelling findings in this manuscript.

OUR RESPONSE: We agree with the reviewer that introducing the analysis of roots with Gibberellin biosynthesis inhibited helps to contextualize our data, and provide insights into potential molecular mechanisms that can concur with the findings reported by Band et al. PNAS 2012. Because our analysis requires single cell file root measurements, we have grown and measured single plant roots treated chemically with Paclobutrazol (PAC), an inhibitor of Gibberellins biosynthesis. We indicate here below the same response to reviewer 1 related to this same topic:

These chemically inhibited roots are known to exhibit the same phenotype as Gibberellin biosynthesis mutants (Band et al, PNAS 2012). In addition, according to reported data on cortex files in Band et al, PNAS 2012, WT roots treated with PAC at concentrations 1uM and 5uM enable to analyse roots with distinct average root growth but the same average mature cortex cell length. This is a scenario we evaluated only computationally in our original manuscript in Fig.3 and which we can now address empirically with these WT plants treated with PAC. Our computational results predicted the existence of a correlation between the number of cells in the EZ and $1/\ln(\text{rez})$ for plants exhibiting these phenotypes.

We have quantified cortex files in single roots treated with these two concentrations of PAC as well as in the WT, following the quantification procedures reported in our original manuscript. Plant materials have been included in Methods (lines 631-635).

The new results are shown in new FIG. 6, and new Appendix Figures S14, S15 and S16. Data in Appendix Fig S14 confirm the reduced root growth and mature cell length previously reported (Band et al, PNAS 2012). We checked these plants follow the same type of exponential profile of cell lengths along the EZ as the WT and performed their automated fitting, as shown in Appendix Fig. S15. Data in Fig.6 and in Appendix Fig. S16 show these roots (for concentrations 1uM PAC and 5uM PAC, respectively) exhibit the correlation between NEZ and $1/\ln(\text{rEZ})$ we predicted computationally.

For completeness, we have also performed numerical simulations for the Ruler, Timer and Sizer models to mimic the WT cortex, as well as that treated with 1uM PAC and with 5uM PAC. These data are shown in FIGS 3 (WT cortex), Fig 6 (WT grown in 1uM PAC) and in Appendix Fig S16 (WT grown in 5uM PAC). The results for the Sizer model are in better agreement with those coming from cortex tissues in WT roots and in WT roots treated with PAC.

These Gibberellin data are presented and discussed in a novel final section of results entitled "The Sizer model is also consistent with the reduction of Gibberellin biosynthesis" (lines 414-443). In the Abstract, Introduction and the Discussion these new results are also mentioned.

In summary, this represents a fascinating manuscript that synthesises modelling simulations and

experimental results as expected for journal such as MSB. Their mechanistic findings are also very interesting and concur well with the findings reported by Band and co-authors in 2012.

OUR RESPONSE: We really appreciate the very positive evaluation of the work and the insights provided to improve it. In the Discussion (lines 515-561) we comment on the results of Band et al. PNAS 2012 and on our new results on GA as well as on bri1-116.

Reviewer #4

Manuscript by Pavelescu et al, is a combined experimental and theoretical study directed for understanding how plant root zonation is established and maintained. Authors specifically focused on how root cells make a decision to differentiate and how this decision is executed in the spatio-temporal context. A very strong point of this study is an integration of quantitative measurements with the theoretical models. Three mechanisms are proposed so called "ruler", "timer" and "sizer" to explain transition from elongation to differentiation with the latter being more fit to the empirical data. Technical aspects of computer modeling are well defined and justified. However, I am not convinced that this study provides novel enough insights in our understanding of root zonation pattern.

1) In my view, all three mechanisms may represent the same concept if one assume that threshold for distance (in ruler model) and timing (in timer model) are not fixed but dynamically changing during root growth as one would expect. In this scenario all three models could be fit to the experimental observations.

OUR RESPONSE: We thank the reviewer for pointing this out. We have now extensively investigated this aspect and provide new theoretical and computational data showing that: - when only the threshold (distance for Ruler, time for Timer and cell length for Sizer) varies from root to root, then all three models become equivalent in terms of which relationships between traits they each predict. We now show that the theoretical prediction on the relationship between the length and the number of cells of the EZ is the same for all models (see section S1.E in Appendix text, and depicted by a blue line in all relevant panels). The computational analysis shows that the sensitivity to the variability in the threshold is distinct between models. In the Timer model, small relative variability in the threshold time (measured as the standard deviation of the threshold time over the mean threshold time) leads to as large effects as a large relative variability in the threshold cell length in the Sizer and in the distance in the Ruler models. In other words, the Timer model is much more sensitive to variability in the threshold than the other models are. These results are in new Appendix Fig. S8 and discussed in the Results section " The three models can give distinct quantitative predictions " (lines 264-268).

- when only the elongation factor or the meristematic activity varies from root to root, then the relationships predicted by each model are distinct and this was the scenario evaluated in our original manuscript (the theoretical ones were described in Appendix Text (section S1E). We have now added new computational data obtained for this scenario, showing it is well described by the theoretical curves, which have been made more precise by taking into account the fact that the number of cells in the EZ is discrete. These new computational data are in new Appendix Fig. S9 (for the three models) and S10 (for the Timer model, showing the different relationship this model has on the cell elongation rate and on the meristematic activity). This is discussed in the Results section "The three models can give distinct quantitative predictions " similarly as it was in the original manuscript (lines 268-316).

- Now we show that all three models can be fine-tuned to drive root growth rates, number of cells in the EZ, length of mature cells, length of EZ and elongation factor consistent with the measured and extracted ones in roots (see more detail in next response). This is shown in new panels B-F of Appendix Figure S4. However, the relationships between phenotypic traits obtained for each model are still slightly different (Fig. 2). The computational data in Figure 2

have been replaced by these new computational data. This point is more detailed in the next answer.

2) Also choice of thresholds for timer and ruler mechanisms is arbitrary and predictions strongly depends on this choice. Would authors provide more comprehensive analysis of average THR in ruler and timer models on the model outcome?

OUR RESPONSE: We agree with the reviewer that a more comprehensive analysis was needed. The threshold for Ruler and Sizer can be chosen based on the direct measurements, yet this is not the case for the Timer model. Therefore, we have searched for parameter values (the threshold mean and its variability, as well as the mean and variability in elongation rate and meristematic activity) in each model that drive outcomes on single phenotypic traits that are consistent with those measured and extracted from the roots described in this manuscript. Five different phenotypic traits were evaluated to assess whether the choice of parameter values was appropriate: 1. root growth rate, 2. number of cells in the EZ, 3. length of mature cells, 4. length of EZ and 5. elongation factor. For each of these five traits, and for each model, Wilcoxon Rank Sum Test was applied to compare the values of the empirical trait and those corresponding to the model (for $n=1000$ and for n equal to the number of measured roots, see new Table EV4). The parameter values for each model were chosen when all tests drive p -values >0.01 .

As expected, for this new choice of parameter values, the three models drive similar outcomes. Yet, each model still predicts slightly distinct relationships between traits.

This procedure has been applied to set the parameter values in all three models for the data of WT roots (for the epidermis and for the cortex, separately), of *bri1-116* and of WT plants chemically inhibited for gibberellin biosynthesis at two different concentrations. These new computational data, together with the empirical ones are presented in new panels in Fig. S4 B-F (for WT epidermis), Figure 3 (WT cortex), Appendix Figure S13 (*bri1-116* mutant), and Fig. 6 and Appendix Figure S16 for WT roots grown in a medium that inhibits gibberellin biosynthesis.

The Results sections "The three models can give distinct quantitative predictions", "The Sizer model is consistent with empirical WT data" and "Brassinosteroid signaling at the meristem is sufficient to modulate the threshold for final cell differentiation" and Discussion have been partially rewritten to present all these new analyses and results.

3) Page 13 paragraph 2, I had hard time to compare data from *bri1-116* mutant and model prediction it is mentioned in the text it fits Sizer model but no actual comparison is made. Also not mechanism is presented to explain how this mutant contributes to reduced size of differentiated cells.

OUR RESPONSE: We are very thankful for this comment. We have searched for parameter values in each model that would correspond to *bri1-116*, following the procedure described in our response to the previous comment. These new computational results are now presented in Fig. S13. We have added cortex data from *bri1-116* roots, which exhibit the same relationships as data from the epidermis (Fig. 5). Pearson correlation coefficient has been evaluated in data coming from each of the three computational models and in *bri1-116* roots (and for all other cases examined). Comparison between model data and *bri1-116* data show that the *bri1-116* data can be explained by a mix of the Sizer and Timer models. These results are presented within the section "Brassinosteroid signaling at the meristem is sufficient to modulate the threshold for final cell differentiation" (lines 387-412)

A hypothetical mechanism for BR contribution in the mechanism mediating the termination of cell elongation is now presented in the Discussion (lines 515-561) and it is accompanied by two Appendix Figs S17 and S18 with new computational data. In this hypothetical scenario a signaling molecule is diluted by cell expansion. Below a threshold concentration, cell elongation ceases. If the molecule is very stable, then a Sizer mechanism arises by sensing this signaling gradient (exemplified in Appendix Fig S17). However, if the molecule is not so stable, then sensing this gradient exhibits the features of a mixed Timer and Sizer mechanisms,

mimicking the change observed between *bri1-116* and the WT (exemplified in Appendix Fig. S18).

List of changes:

- 1- The Abstract has been modified to accommodate for the new results and to fit within the word limit.
- 2- All the main text, Methods and Supporting Information text has been revised to address reviewer's comments. A new final section of results has been created entitled "The Sizer model is consistent with the reduction of Gibberellin biosynthesis".
- 3- The panels in former Figure 2 D-F that had computational and theoretical results have been substituted by new ones, which correspond to new results from new parameter values. For clarity, panel C in former Fig. 2 has been removed. We identified that we were not plotting the measured length of the EZ but instead an inferred one. We have corrected this in all figures and tables that contained these data, plotting now the measured data as indicated below. Specifically, for Fig.2 D with these data has been replotted, substituting the data on the length of EZ by measured values. Panels E,F of experimental data have been stylistically changed but show the same data.
- 4- A new Figure 3 has been created that analyses novel cortex WT data and compares them with new computational and theoretical data of the three models. (Former Fig.3 is now Fig.4).
- 5- In former Fig.4 (now Fig.5) panels C-F with data from the epidermis have been replotted, substituting the data on the length of EZ by measured values (and not inferred ones) and *bri1-116* and *prp5* data on the cortex tissue have been included.
- 6- A new Figure 6 has been created that analyses novel cortex data of WT roots chemically inhibited for gibberellin biosynthesis (paclobutrazol at 1 μ M) and compares them with new computational and theoretical data of the three models.
- 7- Panels A and B in former Fig. S4 have been replaced by new ones, which correspond to new parameter values. In addition, three new panels have been added to extend the type of comparison made between models and epidermal data to three additional phenotypic traits.
- 8- Fig. S8 has been replaced with novel computational data for new parameter values and constitutes now Appendix Figure S9.Boxplots of five different phenotypic traits have been added. The empirical data have been removed, since they are already shown in Fig.2.
- 9- A new Appendix Fig. S9 has been created which sows computational and theoretical results for the three (Ruler, Timer and Sizer) models. These new data evaluate the impact of the threshold value that sets cell elongation cessation.
- 10- Former Figs S9 and S10 and now renamed Appendix Figs. S11 and S12.
- 11- A new Appendix Fig. S10 with new computational data for the Timer model has been created to show its resemblance to the Sizer model under specific conditions.
- 12- A new Appendix Fig. S13 has been created with new computational and theoretical data of the three models to compare them with the data we already had from the epidermis in *bri1-116* mutant (those shown in Fig. 5).
- 13- A new Appendix Fig S14 has been created that shows our new empirical data on Col-0 roots chemically inhibited for the biosynthesis of gibberellin for two different concentrations, together with the WT control, and confirms previous findings by Band et al. PNAS 2012.
- 14- A new Appendix Fig S15 has been created that shows the newly measured cell length profile along single root cortex files for Col-0 roots chemically inhibited for the biosynthesis of gibberellin for two different concentrations, together with the WT control. The automated fitting is also shown as well as the coefficient of determination of the fitting in the elongation zone.
- 15- A new Fig S16 has been created analogous to new Fig 6 but for a higher concentration of paclobutrazol.
- 16- A new Appendix Fig. S17 has been created with new computational data on three novel computational models. These models are based on three different signaling gradients, which drive one type of mechanism (Ruler, Timer, Sizer) each.
- 17- A new Appendix Fig. S18 has been created with new computational data from the new models of signaling gradients, showing mixed (Ruler and Sizer, Timer and Sizer) scenarios.
- 18- To assess the models more quantitatively we have introduced the correlation coefficient (r) in all panels that depict relationships between phenotypic traits (both from computational and from empirical data), substituting the coefficient of determination R^2 that was previously indicated. These values are shown in Figs. 2,3,5,6 and Appendix Figs. S13, S16-S18.

- 19- In former Table S2 (now Table EV2) the p-values have been introduced. The measured values for the length of the EZ have been introduced to substitute inferred ones and statistical comparison between epidermis and cortex for this phenotypic trait has been included.
- 20- In former Table S3 (now Table EV5) the p-values have been introduced. The measured values for the length of the EZ have been introduced to substitute inferred ones and statistical comparison between genotypes and the WT control for this phenotypic trait has been included.
- 21- A new Table EV3 has been created with all parameter values of the computational models. While previously we used a single set of parameter values, now we used 6 sets, to mimic a generic case, the Col-0 epidermis, the Col-0 cortex, *bri1-116* epidermis, Col-0+1 uM PAC cortex and Col-0+5uM cortex. The section of parameter values in Materials and Methods has been modified to accommodate for this.
- 22- A new Table EV4 has been created with all p-values from comparison of the three models with the data from roots, for all types of roots analysed.
- 23- The methods that describe the new experiments and new computations and statistical analyses have been added.
- 24- The simulation code of the automated fitting is now provided as Appendix text -section S3A.
- 25- Raw data of cell length measurements in the stationary regimes will be provided in public database.
- 26- Computational data now depicts the cell in the elongation zone (EZ) next to the differentiation zone (DZ). Previously it depicted the cell in the DZ next to the EZ.
- 27- We have now added in the main text that the mature cell length used in main figures corresponds to the length of the last cell in the EZ, which does not show statistical significant differences (Wilcoxon rank sum test) when compared with the cell with hair (line 257-258, Appendix Fig. S4). This was already indicated in the figure 2 caption in our original manuscript.
- 28- We have now detailed how the samples in which no elongation zone was extracted (1 plant out of 122 WT plants analysed in the epidermis; 1 plant out of 60 WT plants analysed in the cortex and 1 out of 126 in *bri1-116*) were analyzed (lines 681-685).
- 29- In Appendix Text, section S1.E has a new subsection that shows the theoretical predictions on threshold changes. This prediction is plotted in blue line in figures. The theoretical predictions that were in former Section S1.E for changes in the cell elongation rate and the meristematic activity (plotted with gray, black and red lines in Figures) have been extended to describe the cell length of the EZ cell next to the root hair cell and to provide average values of the length of the EZ that take into account that the number of cells in the EZ is discrete (and not a continuous value as assumed in the submitted version). These theoretical mathematical predictions are now compared with the values obtained from the computational models in Appendix Figs. S8-S10. Errors found in formulae (6), (11), (12) and (13) of Section S1B have been corrected.

2nd Editorial Decision

7 November 2017

Thank you again for sending us your revised manuscript. We have now heard back from the two referees who agreed to evaluate your study. As you will see below, the reviewers think that most of the previously raised issues have been satisfactorily addressed. However, reviewer #1 lists a few remaining concerns, which we would ask you to address in a minor revision.

 REVIEWER REPORTS

Reviewer #1:

The authors have extensively revised the manuscript and added a significant amount of new data. They have addressed most of my concerns.

Although the article successfully addresses the points raised by the reviewers, the text remains a bit confusing at times. It would be helpful to add some extra (intuitive) explanations and interpretations of what the equations and measurements tell us. Below are some examples :

- In the introduction and results section the authors state that *bri1* is well described by a mix of the Sizer and Timer models and subsequently conclude 'that BR signaling through BRI1 is required to facilitate the Sizer mechanism.' This is a bit of a shortcut. The conclusion seems counterintuitive, as formally one could also argue that BRI1 is required to facilitate a Timer mechanism (but that would not be in line with the analysis of the wild type). Please explain more clearly what you mean.

- Inhibiting GA synthesis leads to a reduced cortex mature cell length. Yet, (line 443) the authors conclude that « impairment of GA signaling ... does not modify the mechanism that settles cell elongation termination. » Please explain more clearly how shorter final cell length can be obtained without resetting final cell length in the context of a sizer mechanism, which at first sight seems a counterintuitive statement. I guess you mean that a sizer mechanism appears to be still active when GA synthesis is impaired.

- in the discussion ; e.g. line 545 and onwards : « BR signaling stabilization of a signaling component that becomes diluted by cell expansion could underlie the Timer response in *bri1-116*. According to this scenario, when BR signaling is active, the signaling molecule decreases only by dilution, fully mediating a Sizer mechanism, whereas it becomes degraded in the absence of BR signaling, becoming a signature of time as well. » In a context where the sizer mechanisms is put forward this formulation is confusing. Please try to be as clear as possible.

Some details :

The timer mechanism as defined here might also depend on the temperature (cf the term 'thermal time' used in many ecophysiological models). This is of course of no consequence here, as all experiments are carried out at the same temperature.

The models mainly take into account cell length and not volume. The cells in the *bri1* mutant, for example are shorter, but also wider. Til what extent would this modify the outcome of the analysis ?

Line 28 : 'how often the meristem divides' should be 'how often the meristematic cells divide'. Note that it is not only division, but also growth. Probably the term proliferation would cover that better (?)

Reviewer #4:

I went through the revised manuscript and I conclude that authors did a great job to improve overall quality of the manuscript and answer satisfactory to my comments. Therefore, I now recommend this manuscript for publication in *Molecular Systems Biology*.

2nd Revision - authors' response

21 November 2017

POINT BY POINT RESPONSE:

Reviewer #1:

The authors have extensively revised the manuscript and added a significant amount of new data. They have addressed most of my concerns.

Although the article successfully addresses the points raised by the reviewers, the text remains a bit confusing at times. It would be helpful to add some extra (intuitive) explanations and interpretations of what the equations and measurements tell us. Below are some examples :

- In the introduction and results section the authors state that *bri1* is well described by a mix of the Sizer and Timer models and subsequently conclude 'that BR signaling through BRI1 is required to facilitate the Sizer mechanism.' This is a bit of a shortcut. The conclusion seems counterintuitive, as formally one could also argue that BRI1 is required to facilitate a Timer mechanism (but that would

not be in line with the analysis of the wild type). Please explain more clearly what you mean.

Our answer: We thank the reviewer for pointing out distinct points that deserve further explanation and clarification in the text.

In the Introduction, we now clarify the role of BRI1 by stating (lines 119-128):

" To evaluate further this mechanism we analysed roots with reduced mature cell lengths, such as the BR insensitive mutant *bri1-116*. Our analysis supports that the Sizer model, with reduced threshold length, cell elongation rate and meristematic proliferation, is not sufficient to account for the quantitative data in *bri1-116*. Instead, this mutant is well described by a mix of the Sizer and Timer models. This suggests that BR signaling through BRI1 suppresses the Timer mechanism, which appears to participate in the absence of BRI1-mediated signaling and not in the WT. Thereby, BRI1 signaling facilitates that the termination of elongation proceeds only through the Sizer mechanism in the WT, while increasing the threshold length, cell elongation rate and meristematic activity."

In the Results section we have modified lines 393-409 to clarify this aspect. It now reads:

" To test which elongation termination mechanism is present in the *bri1-116* mutant, we focused on the phenotypic traits 'number of cells in EZ' (NEZ), 'length of EZ' (LEZ), 'elongation factor' (rEZ) in single roots in the stationary regime (n=126 for epidermis, corresponding to day 6, 8 and 10 post germination, Dataset EV2) and on whether the three models could reproduce them. As expected (see Introduction), all three models can account for the change observed in each phenotypic trait of *bri1-116* (Appendix Fig. S13, Table EV4). All models involve a reduced meristematic activity and relative cell elongation rate compared to the WT, but only the Ruler and Sizer models have a reduced threshold of differentiation (Table EV3). As done for the WT data, we then assessed whether the relationships between these traits are best described by any of the models (Fig. 5E-H). In contrast to the WT, the correlations between these traits in the *bri1-116* mutant exhibit features of both the Timer and Sizer models (Fig. 5E-H, Appendix Fig. S13). This suggests that cell elongation termination in the *bri1-116* mutant proceeds through a mixed mechanism between the Sizer and the Timer. Therefore, the absence of BRI1 signaling enables the Timer mechanism to take a role, which is not apparent in the WT. This suggests that BR signaling through BRI1 facilitates the Sizer mechanism to dominate in the WT. "

Lines 424-427 have also been changed to:

" Together, our results suggest that BRs are sufficient at the meristem to restore root growth and the mature cell length and that BR signaling is required to make the Sizer mechanism the dominant one. "

Reviewer #1:

- Inhibiting GA synthesis leads to a reduced cortex mature cell length. Yet, (line 443) the authors conclude that « impairment of GA signaling ... does not modify the mechanism that settles cell elongation termination. » Please explain more clearly how shorter final cell length can be obtained without resetting final cell length in the context of a sizer mechanism, which at first sight seems a counterintuitive statement. I guess you mean that a sizer mechanism appears to be still active when GA synthesis is impaired.

Our answer:

In the Introduction we have clarified now this point by stating (lines 130-136):

"Finally, we show that the growth of plants chemically inhibited for the biosynthesis of gibberellin, which are known to have short mature cell lengths (Band et al, 2012; Ubeda-Tomás et al, 2009, 2008), is consistent with the Sizer mechanism, like in the WT, but with decreased threshold length, elongation rate and meristematic activity. Therefore, the results suggest that both gibberellin and BR signaling, which are known to crosstalk (Ross & Quittenden, 2016), participate in setting cell expansion termination, although in very distinct manners. "

In the results section we have also clarified this aspect as follows (lines 468-472):

"These results indicate that impairment of GA signaling, in contrast with impairment of BR signaling, does not change which is the mechanism that settles cell elongation termination. Our data support the Sizer mechanism as the one taking place when GA biosynthesis is impaired, as in the WT. Yet, GA biosynthesis impinges on cell elongation termination by modifying the threshold length."

In addition, the text in this section (lines 444-466) has been partially reordered and rephrased to make it more explanatory and to cite the panels of Fig.6 in the order they appear in the figure. These lines (444-466) read:

" Our analysis in Fig. 4 indicates that the correlation between the number of cells in the EZ (NEZ) and the spatial increase in cell length along the root (as measured by $1/\ln(rEZ)$) is relevant to account for roots with equal mature cell length, but distinct root growth. The same analysis predicts that roots treated chemically at these two different concentrations will both exhibit such a correlation.

As done with the WT and *bri1-116* mutant, all three models could be fitted to adjust to single phenotypic traits of root growth in the roots grown under the two different concentrations of PAC (Fig. 6A for $1\mu\text{M}$ PAC and Appendix Fig. S16 for $5\mu\text{M}$ PAC, Table EV4). All models involved a change in the differentiation threshold, the relative cell elongation rate and the meristematic activity, compared to the WT (Table EV3). We then evaluated the three relationships between the phenotypic traits of l_{max} , LEZ, NEZ and rEZ in these chemically treated plants roots (Fig. 6B-D for $1\mu\text{M}$ PAC and Appendix Fig. S16 for $5\mu\text{M}$ PAC, Dataset EV2). As predicted from our computational analysis in Fig.4, NEZ is correlated with $1/\ln rEZ$ in these plants (Fig. 6D, Appendix Fig. S15). In addition, the three relationships are of the same type as in the WT (Fig. 6B-D and Appendix Fig. S16), suggesting that the Sizer mechanism drives cell elongation termination in these roots. Comparison of these relationships with those arising from each model confirms that the Sizer model describes best the correlations between these traits (Figs. 6B-D and Appendix Fig. S16). While the Ruler model is not able to drive such relationships, the Timer model would only drive them when the variability in the threshold time and in the relative cell elongation rate are assumed to be small enough, which is not expected to happen based on our quantification of dynamical traits."

The title of this section of Results has also been changed from "The Sizer model is also consistent with the reduction of Gibberelin biosynthesis" to " The Sizer model is also consistent with data on roots with reduced Gibberelin biosynthesis" for clarity (lines 429-430).

Reviewer #1:

- in the discussion ; e.g. line 545 and onwards : « BR signaling stabilization of a signaling component that becomes diluted by cell expansion could underlie the Timer response in *bri1-116*. According to this scenario, when BR signaling is active, the signaling molecule decreases only by dilution, fully mediating a Sizer mechanism, whereas it becomes degraded in the absence of BR signaling, becoming a signature of time as well. » In a context where the sizer mechanisms is put forward this formulation is confusing. Please try to be as clear as possible.

Our answer:

The explanation on gradients and their relationship with the Sizer mechanism has been clarified, as well as the potential role of BR on them, such that the paragraph in the Discussion from lines 544-594 has been partially rewritten and now reads:

"Yet, our results suggest that the Timer mechanism is involved in root growth in the absence of BRI activity. Moreover, the analysis of similar short root phenotypes that are driven by the alteration of a distinct hormone signaling (i.e. GA signaling) show this role of BRI1 is specific. Hormone signaling gradients can be expected to mediate Ruler, Timer or Sizer mechanisms depending on how the gradient is formed (Appendix Fig. S17). For instance, when diffusion and degradation drive a spatial signaling gradient, the concentration of the signaling molecules depends on their spatial position relative to where they were produced (Crick,

1970). Thus, sensing this signaling gradient can provide positional information and mediate a Ruler mechanism (see Appendix Fig. S17 for simulations). In contrast, a signaling gradient formed only by dilution within cells expanding and becoming displaced is dictated by the cell length (Band et al, 2012) and thereby can mediate a Sizer mechanism (Appendix Fig. S17). For instance, a Sizer mechanism arises when cell elongation terminates once the concentration of this signaling molecule within the cell is below a threshold value (Appendix Fig. S17). Computational and mathematical modeling has previously proposed that despite being diffused, gibberellin is synthesized mainly at the meristem and its concentration decays across the EZ mostly through dilution (Band et al, 2012). Hence GA concentration across the EZ is dictated by the cell length and not by the distance from the meristem and therefore can potentially mediate the Sizer mechanism. Then, the GA concentration would be below threshold in shorter cells than in the WT when GA biosynthesis is inhibited (but not completely blocked). Thus, reduced GA biosynthesis at the meristem would drive a reduced mature cell length (besides changes in relative cell elongation rate and meristematic activity) and roots should still exhibit the features of the Sizer mechanism, as we find for roots treated with 1 μ M PAC. However, additional reduction of biosynthesis, by higher PAC concentrations, would be expected to drive shorter mature cell lengths (unless additional assumptions are made), in contrast with what is found and suggesting that the GA gradient may not underlie the Sizer mechanism of cell elongation termination in roots. Moreover, recent visualization of the GA gradient in roots challenges the GA gradient itself (Rizza et al, 2017). Alternatively, GA signaling could participate in cell length sensing by modulating BR signaling components through their crosstalk. Crosstalk between BRs and GAs can occur at the level of signaling, such that BR signaling components downstream BRI1 receptor (such as BZR) interact with growth repressors DELLA proteins, which are inhibited by GA, and/or at the level of GA biosynthesis (Ross & Quittenden, 2016). Moreover, we show that BRI1 signaling at the meristem is sufficient for root growth. BRI1 expression under another promoter located at the meristem, RCH1, in the *bri1* mutants partially rescued the wild type phenotype (Hacham et al, 2011). In addition, our results suggest that BRI1 signaling is required for the Sizer mechanism to be dominant, and that the Timer mechanism plays a role, together with the Sizer, in its absence. A potential scenario can be envisaged to account for these results based on signaling gradients across the EZ formed by dilution in expanding cells and in cell elongation termination below a threshold signaling concentration (Appendix Fig. S17). If such a signaling molecule becomes degraded in the absence of BRI1 signaling, and not only diluted by cell expansion, then its concentration within the cells depends on time and on the cell size. Therefore, sensing this molecule for terminating cell elongation would result in a mixed Timer and Sizer mechanism (Appendix Fig. S18). Instead, if the molecule is very stable and is not degraded when BRI1 signaling is present, it only mediates the Sizer mechanism since its concentration depends only on the cell size (Appendix Fig. S17). Changes in the anisotropic cell growth as well as the effect of temperature on the Timer response could be included and evaluated. Yet, other or additional molecular mechanisms could be driving the sensing of the cell length mediated by BR signaling. "

Reviewer #1:

Some details :

The timer mechanism as defined here might also depend on the temperature (cf the term 'thermal time' used in many ecophysiological models). This is of course of no consequence here, as all experiments are carried out at the same temperature.

Our answer: The reviewer is correct. Temperature changes could drive changes in the Timer. The experiments have been performed at a fixed temperature as indicated in Methods. The sentence " Changes in the anisotropic cell growth as well as the effect of temperature on the Timer response could be included and evaluated. " has been added in the Discussion (lines 591-593).

Reviewer #1:

The models mainly take into account cell length and not volume. The cells in the *bri1* mutant, for example are shorter, but also wider. Til what extent would this modify the outcome of the analysis ?

Our answer: This would not modify any of the outcomes of the quantitative analysis nor the fitting to one or other mechanism. This is because the Ruler, Timer and Sizer models described in Results and Methods are independent of how the Distance, Time and Length, respectively, are sensed.

In contrast, this would have an effect, only at the quantitative level, on the models of signaling gradients formed by a molecule that becomes diluted by cell expansion that are discussed in the Discussion (Appendix Figs S17 and S18). These models assume how the cell length is sensed: by measuring the concentration of a signaling molecule. Taking into account the volume, instead of just the cell length, together with the fact that cells widen in *bri116* mutant, would drive the same conclusions. The only difference would be that, for the same parameter values (which are arbitrary), a shorter mature cell length will be predicted.

Since this aspect can be also of interest to be studied, the sentence " Changes in the anisotropic cell growth as well as the effect of temperature on the Timer response could be included and evaluated. " has been added in the Discussion (lines 591-593).

Reviewer #1:

Line 28 : 'how often the meristem divides' should be 'how often the meristematic cells divide'. Note that it is not only division, but also growth. Probably the term proliferation would cover that better (?)

Our answer: We have changed the sentence "how often the meristem divides" for "how often the meristem proliferates" (line 28).

Reviewer #4:

I went through the revised manuscript and I conclude that authors did a great job to improve overall quality of the manuscript and answer satisfactory to my comments. Therefore, I now recommend this manuscript for publication in Molecular Systems Biology

Our response: We appreciate the positive evaluation of the reviewer.

Corresponding Author Name: Ana I. Caño-Delgado & Marta Ibañes

Manuscript Number: 7687